# Hominin-specific regulatory elements selectively emerged in oligodendrocytes and are disrupted in autism patients

Bas Castelijns [1], Mirna L. Baak[1], Ilia S. Timpanaro[1], Caroline R.M. Wiggers[1,2], Marit W. Vermunt[1], Peng Shang[1], Ivanela Kondova [3], Geert Geeven[1], Valerio Bianchi [1], Wouter de Laat [1], Niels Geijsen[1] & Menno P. Creyghton[1,4]*

Speciation is associated with substantial rewiring of the regulatory circuitry underlying the expression of genes. Determining which changes are relevant and underlie the emergence of the human brain or its unique susceptibility to neural disease has been challenging. Here we annotate changes to gene regulatory elements (GREs) at cell type resolution in the brains of multiple primate species spanning most of primate evolution. We identify a unique set of regulatory elements that emerged in hominins prior to the separation of humans and chimpanzees. We demonstrate that these hominin gains perferentially affect oligodendrocyte function postnatally and are preferentially affected in the brains of autism patients. This preference is also observed for human-specific GREs suggesting this system is under continued selective pressure. Our data provide a roadmap of regulatory rewiring across primate evolution providing insight into the genomic changes that underlie the emergence of the brain and its susceptibility to neural disease.

---

[1] Hubrecht Institute-KNAW & University Medical Center Utrecht, Uppsalalaan 8, 3584 CT Utrecht, The Netherlands. [2] Division of Pediatrics, University Medical Center Utrecht, Heidelberglaan 100, 3584 XC Utrecht, The Netherlands. [3] Biomedical Primate Research Center, Lange Kleiweg 161, 2288 GJ Rijswijk, The Netherlands. [4] Department of Developmental Biology, Erasmus University Medical Center, Wytemaweg 80, 3015 CN Rotterdam, The Netherlands. *email: m.creyghton@erasmusmc.nl

Understanding the emergence of the human brain and the unique properties that define our species in evolutionary history remains a major challenge[1]. Furthermore, as several neuropsychiatric and neurodegenerative diseases are suspected to be linked to genetic changes that recently evolved[2–5], unraveling evolution of the human brain may have consequences beyond a plain understanding of the human condition. However, as the human brain is the result of a process that spans the entirety of primate evolution, giving rise to primate brains of variable sizes and cognitive complexities[1,6], its understanding may also require a broader evolutionary context. This is especially relevant given the absence of pervasive data on neural disorders in great apes[7] and the challenges in assessing their cognitive abilities[8].

The process underlying brain development is controlled by gene expression programs that dictate cellular identity in a spatio-temporal manner[9,10]. Gene regulatory elements (GREs) such as enhancers and promoters function as transcriptional units that recruit specific transcription factor complexes to control the expression of genes in a cell-type-dependent manner[11]. This cell specificity as well as functional redundancy of enhancers is likely linked to an enhanced evolutionary flexibility, avoiding some of the pleiotropic effects associated with changes in genes[12]. As such, pervasive rewiring of the gene regulatory circuitry has been observed across mammalian evolution while genes remain mostly conserved[13]. This rewiring includes a host of regulatory changes that were specific to the human brain[14–16]. Despite these efforts, finding a cause—consequence relationship between important evolutionary alterations in gene regulatory elements, evolution of the brain and its susceptibility to neural disorders has remained difficult. Here we annotate regulatory changes in the brain across primate evolution. We identify a set of regulatory changes that emerged after the separation from old world monkeys but prior to the separation between chimpanzee and human. These elements are referred to as hominin-specific and are preferentially enriched in oligodendrocytes in adults and deregulated in the brains of autism spectrum disorder (ASD) patients. We propose that evolution of regulatory DNA in hominins may have helped set the stage for the emergence of the human brain and its susceptibility to disease.

## Results

**Annotation of hominin-specific regulatory change in brain.** To track relevant genome changes in the brain across primate evolution, we annotated GREs in brain tissue from three healthy marmoset specimens (Supplementary Data 1). With a common ancestor living approximately 44 million years ago[17], the divergence of marmosets and humans spans a major part of primate evolution (Fig. 1a). We analyzed prefrontal cortex (PFC), a key anatomical region involved in many of the executive processes that define our species and cerebellum (CB), a neuron dense structure harboring mainly granule neurons[18]. To identify active GREs, we used ChIP-Seq for histone H3 lysine 27 acetylation (H3K27ac), a robust assay to identify active GREs on a global scale[19]. In addition, we analyzed trimethylation of histone H3 lysine 4 (H3K4me3), which is specifically associated with active transcriptional start sites (TSS)[20] (Supplementary Data 3). Data were well within quality standards[21] and were reproducible between biological replicates ($r > 0.9$, Supplementary Fig. 1a, b). We identified 33,754 marmoset GREs of which 16,086 were predicted promoter GREs and 17,668 predicted enhancer GREs (Supplementary Fig. 1c). Using human annotated promoter sequences, we observed that the majority (75%) of predicted promoters in marmoset overlapped with annotated promoters in humans. Consistent with previous observations, analyses of RNA-

Seq data confirmed that active GREs were more often associated with active gene expression in marmoset brain (Supplementary Fig. 1d)[22,23].

To assess regulatory changes across primate evolution, we focused on H3K27ac enrichment and compared our data to active GREs identified in rhesus macaque, chimpanzee and human in PFC and CB (Supplementary Fig. 2a, b)[14]. Only GREs that could be consistently mapped onto all four genomes were included in the analyses (Supplementary Fig. 2c—e). While this excludes species-specific DNA, most GREs that were excluded were discarded due to poor genome annotation and/or ambiguous mapping of reads. This is consistent with the observation that regulatory changes primarily occur in conserved DNA as opposed to DNA that is evolutionary novel[13]. We retained a total of 37,308 GREs that could be mapped on all four species of which 25% overlapped a TSS in humans (Supplementary Data 4, 5). Dimension reduction and visualization with t-SNE and hierarchical clustering revealed a clear separation of the two anatomical locations as well as the major primate clades, with high correlation between replicate samples (Fig. 1b, c, Supplementary Fig. 2f).

While a prior analysis focused on identifying regulatory changes specific to the human brain[14], significant differences in brain size as well as the emergence of complex behavior have also occurred prior to the separation of humans and chimpanzee in great apes[1,6]. To gain insight into the regulatory changes occurring prior to human evolution, we first selected elements that were differentially enriched between human and both marmoset and rhesus macaque using DESeq2 as demonstrated previously[14] (Supplementary Fig. 3a). The same analysis was performed using chimpanzee data instead of human data and the resulting datasets were compared (Supplementary Fig. 3b). Similar to our prior analysis[14], biological replicates were generated in separate batches to ameliorate batch-related effects and no major batch effects were observed (Supplementary Fig. 3c—e). We found 1398 (713 CB, 685 PFC) regions that were designated as hominin (humans and chimpanzee)-specific gains and 532 (374 CB, 158 PFC) that were defined as hominin-specific losses (Fig. 1d, Supplementary Fig. 4a, Supplementary Data 6). For instance, several hominin-specific regulatory changes were found close to the *SORCS1* gene in CB (Supplementary Fig. 4b), which is a known regulator of synaptic trafficking and linked to aggression[24,25]. Mutations in this gene have been linked to Alzheimer's disease (AD) as well as ASD[25]. The observed imbalance between the hominin-specific gains and losses observed is likely due to hominin losses as defined here require conservation of activity across a longer evolutionary period (i.e. conservation between rhesus macaque and marmoset). As such, these conserved regions are more likely to be biologically relevant and thus their loss may be selected against. Hominin-specific gains were less frequently shared with other tissues (Supplementary Fig. 4c), consistent with previous observations suggesting that evolutionary changes occur preferentially in tissue-specific GREs[14]. They were also less frequently associated with promoters consistent with an enhanced evolutionary flexibility at distal enhancers (Supplementary Data 5)[13]. More regions were found differentially enriched in CB compared to PFC that is consistent with previous ChIP and gene expression analysis[14,23] and likely due to CB being more homogeneous resulting in better resolved GREs that are easier to compare. Motif analysis did not yield a clear enrichment for transcription factor binding sites at these elements, suggesting their change was not directly linked to a single transcriptional program that was altered (Supplementary Data 7). This was further supported by an overall reduction in primate sequence conservation at these elements compared to elements that did not change activity during primate evolution

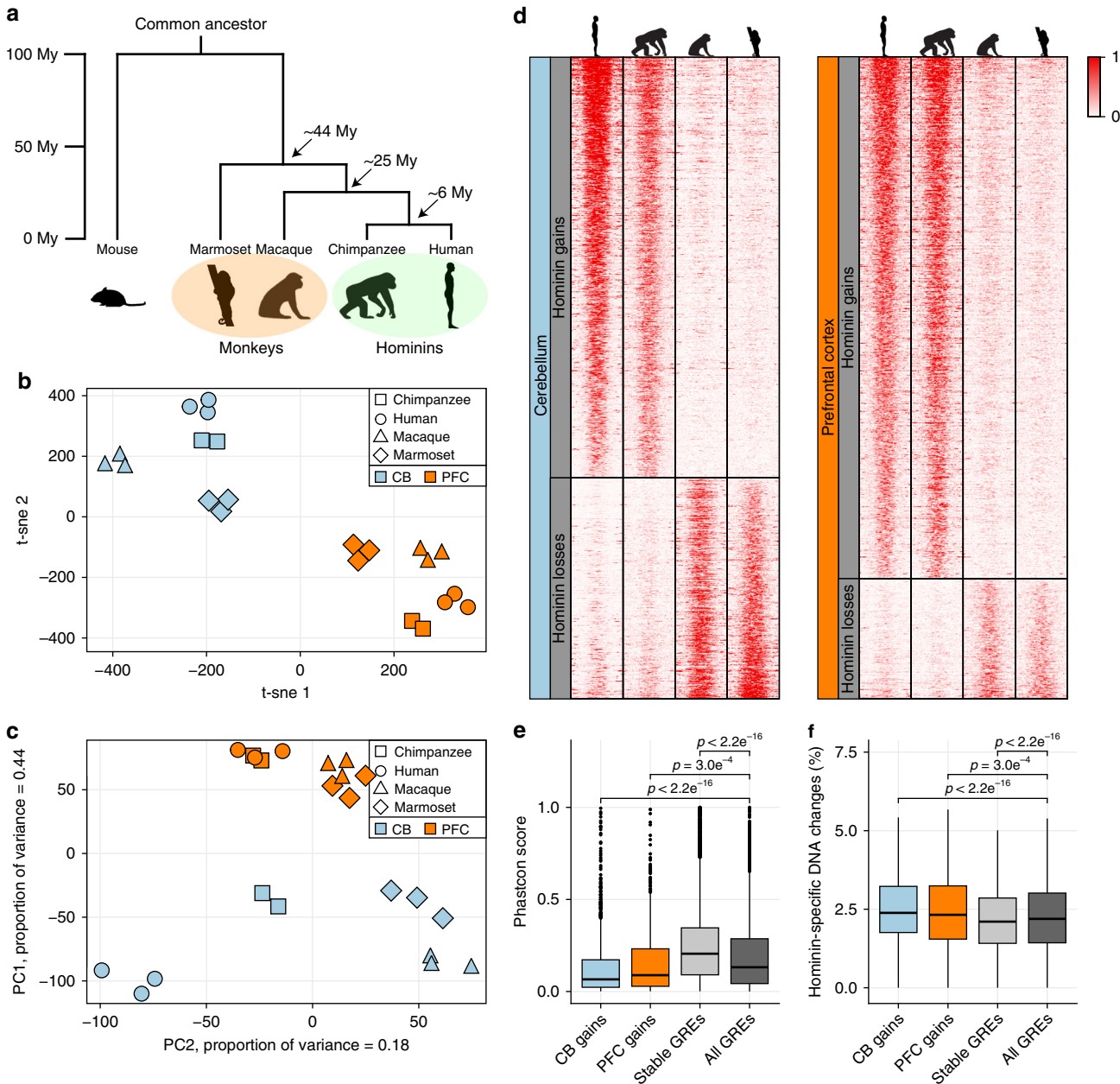

**Fig. 1 Identification of hominin-specific regulatory changes in brain tissue. a** Schematic representation of the primate phylogenetic tree with time indication of the major branch points. **b** t-Distributed Stochastic Neighbor Embedding (t-sne) analysis of all H3K27ac-enriched GREs with orthologs on all four primate genomes ($n = 37,308$). Axes indicate semantic space (CB cerebellum, PFC prefrontal cortex). **c** PCA analysis of the same regions as in (**b**), shown for the first two principal components. **d** Heatmap showing H3K27ac enrichment on scaled hominin-specific regulatory changes in both cerebellum and prefrontal cortex. Heatmap colors indicate H3K27ac enrichment (rpm). CB gains $n = 713$; CB losses $n = 374$; PFC gains $n = 685$; PFC losses $n = 158$. **e** Conservation scores (20 mammals) using PhastCon as defined by UCSC for hominin-specific gains and evolutionary stable GREs compared to all here identified GREs. Dissimilarities between distributions were calculated using a Student's t test. **f** Analysis as in (**e**) for hominin-specific nucleotide changes. Bottom and top of the box plots are the first and third quartile. The line within the boxes represents the median and whiskers denote interval within 1.5× the interquartile range from the median, outliers are depicted as points. Source data are provided in Source Data file.

(Fig. 1e). Furthermore, we observed an increase of hominin-specific nucleotide changes at these GREs, suggesting that the observed regulatory changes are sequence based (Fig. 1f). Thus, our analysis has uncovered a unique set of regulatory elements that were conserved in monkeys but changed activity in hominins.

**Hominin PFC gains preferentially emerged in oligoden-drocytes.** As the cortex is highly heterogeneous, containing a variety of neural and glial cell types, we asked whether GREs that

recently evolved in hominins were spatially confined to the frontal cortex or rather restricted to a particular cell type. We therefore analyzed single-cell ATAC-Seq data generated in human PFC[26], and found that hominin-specific gains were overwhelmingly overrepresented in oligodendrocyte-specific open chromatin domains in PFC (Fig. 2a). To independently confirm this observation, we analyzed H3K27ac data derived from FANS sorted NeuN+ glutamatergic neurons, Sox6+ GABAergic neurons and Sox10+ oligodendrocytes[27], FACS sorted CD11b+ CD45low CD64+ CXCR1high microglia[28] as well as primary astrocytes

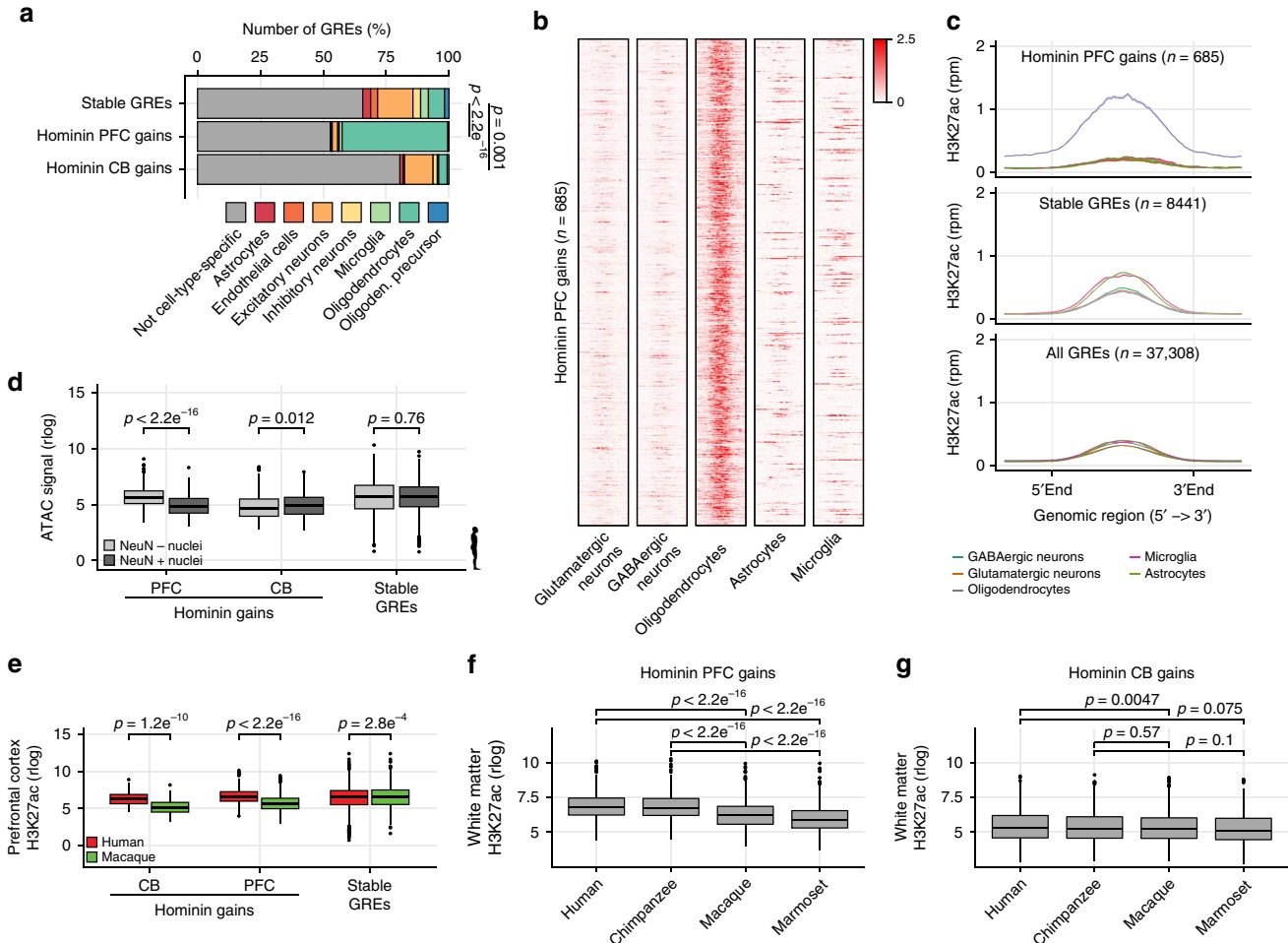

**Fig. 2 Hominin-specific gains are oligodendrocyte-specific. a** Bar plot showing cell-type-specific ATAC peaks in hominin-specific gains in PFC and CB and stable GREs. Difference in oligodendrocyte-specific open chromatin frequency was calculated using a Fisher's exact test. **b** Heatmap showing H3K27ac enrichment for 685 scaled hominin-specific PFC gains, analyzed in different cell types as indicated. Heatmap colors indicate H3K27ac enrichment (rpm). **c** Metaplot analysis showing the average H3K27ac enrichment profile in different cell types for hominin-specific PFC gains, stable and all GREs. **d** Box plot showing normalized ATAC signal for hominin-specific gains and stable GREs in nuclei FANS sorted for NeuN. Dissimilarities between distributions were calculated using a Student's *t* test. **e** Box plots showing normalized H3K27ac enrichment for hominin-specific gains and stable GREs in prefrontal cortex in both human and rhesus macaque. Dissimilarities between distributions were calculated using a Student's *t* test. **f** Box plots showing normalized H3K27ac enrichment for hominin-specific PFC gains in white matter tissue per species. Dissimilarity between distributions was calculated using a Student's *t* test. **g** Box plots as in (**f**) but for normalized H3K27ac enrichment on hominin-specific CB gains. Bottom and top of all box plots are the first and third quartile. The line within the boxes represents the median and whiskers denote interval within 1.5× the interquartile range from the median, outlier are depicted as points. Source data are provided in Source Data file.

isolated from adult brain[29] (Supplementary Fig. 5a). We found that most of the GREs that evolved in the PFC of hominins were selectively enriched in Sox10[+] oligodendrocytes and not in other neural or glial cell types (Fig. 2b, c, Supplementary Fig. 5b). Hominin-specific gains in cerebellum did not show strong enrichment for cell-type-specific regions consistent with granule neurons not being represented by these data (Supplementary Fig. 5c). As oligodendrocytes represent the main constituent of glial cells in the brain, we used a third independent dataset[30] separating neurons and glial cells in PFC based on the expression of NeuN which selectively labels neural nuclei on the nuclear membrane and not glial nuclei. We confirmed that hominin-specific gains in PFC were selectively enriched in glial cells (Supplementary Fig. 5d, e).

As a modest increase in glia to neuron ratio, scaling with brain volume, was observed in larger primates[31–33], we wondered whether the link between regulatory gains and oligodendrocytes was the result of an altered glial content in the cortex. To address this, we generated ATAC-Seq data from human, chimpanzee, rhesus macaque and marmoset frontal cortex resolved between neuronal and glial cells by FANS sorting for NeuN (Supplementary Fig. 6a−c, Supplementary Data 3). Consistent with cell-type-specific H3K27ac enrichment and the single-cell ATAC-Seq data[34], hominin-specific PFC gains were enriched for open chromatin in NeuN− nuclei compared to NeuN+ nuclei in humans (Fig. 2d). The opposite was observed for hominin-specific CB gains, which is consistent with an overwhelming neural content in CB (Fig. 2d). In addition, we found that ATAC-Seq signal at hominin-specific PFC gains was enhanced in humans and chimpanzee only in NeuN− nuclei and not in NeuN+ nuclei (Supplementary Fig. 6d). Moreover, we observed a slight increase in H3K27ac enrichment in regions specific to NeuN− nuclei in humans compared to rhesus macaque; however, this was substantially less pronounced than the differences observed for the regions identified as hominin-specific gains (Supplementary Fig. 6e). This demonstrates that regions that gain

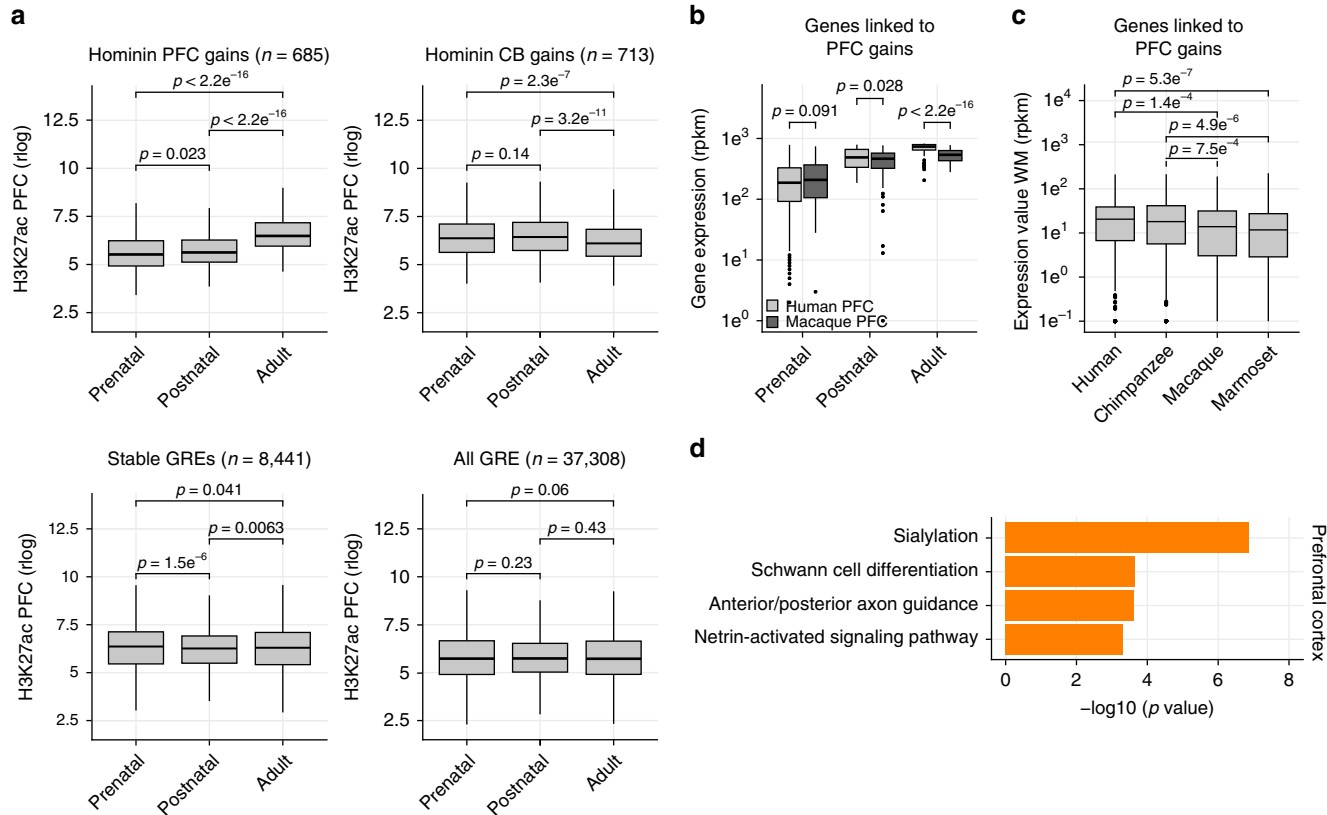

**Fig. 3 Hominin-specific gains emerge postnatally. a** Box plots showing normalized H3K27ac enrichment in PFC or CB samples across different developmental timepoints for hominin-specific gains, stable GREs and all primate GREs. Dissimilarities between distributions were calculated using a Student's $t$ test. **b** Box plots showing normalized gene expression values of genes linked to hominin-specific PFC gains across developmental time, for both human and rhesus macaque. Dissimilarities between distributions were calculated using a Student's $t$ test. **c** Box plots showing normalized gene expression values in the white matter tissue of different primates for genes linked to hominin-specific PFC changes. Dissimilarities between distributions were calculated using a Student's $t$ test. **d** Gene-ontology analysis using GREAT for hominin-specific PFC gains. $P$ values represent significance of enrichment for stated biological process. Bottom and top of all box plots are the first and third quartile. The line within the boxes represents the median and whiskers denote interval within 1.5× the interquartile range from the median, outliers are depicted as points.

H3K27ac enrichment in hominins are also preferentially in an open configuration in hominins. Furthermore, these data suggest that the skewing of hominin-specific gains towards oligodendrocytes is not solely due to an altered neural to glial ratio. To further exclude altered glial content as a factor in our analysis, we generated and analyzed H3K27ac ChIP-Seq data in white matter (WM) for all four primate species (Supplementary Fig. 6f–g, Supplementary Data 3). WM lacks neural content and consists primarily (~75%) of oligodendrocytes[32]. Hominin-specific gains in PFC also gained of activity in human and chimpanzee WM compared to rhesus and marmoset, further excluding glial content as the main factor in our analysis (Fig. 2f). Hominin-specific gains identified in CB were not differentially enriched in WM (Fig. 2g). This strongly argues against these results emerging from an altered neural to glial ratio in PFC. Furthermore, it also suggests that the differences at these GREs are not spatially confined to the PFC. Thus, our data demonstrate that hominin-specific regulatory changes associate with hominin-specific sequence changes, preferentially affecting GREs in oligodendrocytes.

**Hominin gains activate postnatally and link to myelination.** To analyze whether there was temporal skewing of hominin-specific gains towards a particular developmental stage, we analyzed H3K27ac data from distinct developmental timepoints covering prenatal, postnatal and adult stages in humans[35]. We found that hominin-specific PFC gains preferentially associate with regulatory

DNA that activates between the postnatal and adult stages (Fig. 3a, Supplementary Fig. 7a) which, given their enrichment in oligo-dendrocytes, is consistent with increased myelination during this developmental stage[1]. This was not due to our initial analysis being limited to adult tissue, as GREs identified here were overall not differentially enriched between the postnatal and adult stages (Fig. 3a).

To link hominin-specific gains to their target genes, we used a combination of proximity-based gene linkage as well as HiC-derived enhancer–promoters links from analysis in PFC[36]. We used promoter locations based on the human genome to link GREs to genes for all primates as it the best annotated genome and promoter locations are largely conserved throughout primate evolution (Supplementary Fig. 7b, c). Using independent RNA-Seq data generated in PFC and CB for humans and rhesus macaque[15,23], we found that the expression changes of genes linked to hominin-specific gains correlated with H3K27ac gains in CB and PFC in humans (Supplementary Fig. 7d). In addition, we observed that these gene expression differences also predominantly occurred postnatally in both PFC and CB (Fig. 3b, Supplementary Fig. 7e) supporting the postnatal selectivity of hominin-specific gains. To verify that the expression changes for genes linked to hominin-specific PFC gains were consistent in glial cells, further excluding changes in glia to neural content as a factor in these observations, we generated WM RNA-Seq data from the four primate species (Supplementary Fig. 7f, g, Supplementary Data 3). In agreement with the changes at GREs,

hominin-specific gains at regulatory DNA were mirrored by gene expression changes in WM (Fig. 3c, Supplementary Fig. 7h).

Functional analysis of genes linked to hominin-specific PFC changes revealed that they were enriched for genes involved Schwann cell differentiation, a myelinating cell of the nervous system further supporting oligodendrocyte identity (Fig. 3d, Supplementary Data 8). Furthermore, we found hominin-specific H3K27ac enrichment near genes involved in the regulation of axon guidance and sialylation while this was not observed for hominin-specific gains in cerebellum (Fig. 3d, Supplementary Fig. 7i). For instance, hominin-specific PFC gains were located near the sialyltransferase *ST3GAL5* (Supplementary Fig. 7j), which causes intellectual disability when mutated[37]. Sialic acid is a key monosaccharide for the synthesis of brain gangliosides and sialylated glycoproteins (NCAMS), both of which are essential for cognitive stability and brain development[38]. While sialic acid is especially enriched in brain tissue, it is also found in high concentrations in human milk and believed to play a role in postnatal synaptic plasticity and memory formation[39]. Indeed, defects in these biosynthesis pathways have been linked to axon instability, altered nerve excitability and psychiatric disorders including schizophrenia and ASD[38]. This may suggest that some regulatory features of these disorders link to evolutionary changes that emerged in great apes and predates the separation of humans and chimpanzees.

**PFC hominin gains are deregulated in ASD patient brain**. As a link between the evolution of the human brain and the emergence of several neural disorders was proposed previously[2–5], we further explored the potential link between regulatory changes emerging in hominins and disorders of the brain. We first compared hominin-specific regulatory changes to genome-wide association (GWAS) data, to link nucleotide variants associated with AD[40], ASD[41], Parkinson's disease[42], bipolar disorder and schizophrenia[43] using stratified LD score regression on the summary statistics of these variants[43]. While enrichment for heritability to schizophrenia was observed at GREs, particularly those that were evolutionary stable and promoters (Supplementary Fig. 8a), we did not observe a significant enrichment for disease-associated variants in GREs that were gained or lost in hominins (Supplementary Fig. 8a, b).

To contrast our data to more direct evidence for GRE deregulation in neural disease, we also analyzed H3K27ac ChIP-Seq data across 45 autism patient brains and 49 control samples, similarly covering PFC and CB[44] as well as H3K27ac ChIP-seq data from the entorhinal cortex of 47 Alzheimer's disease patients[45]. In agreement with previous analysis we found a correlation between regions that were deregulated in ASD patients and GWAS variants linked to schizophrenia (Supplementary Fig. 8a). Surprisingly, we also found strong evidence for a link between hominin-specific regulatory gains in the PFC and regions that lose H3K27ac enrichment specifically in ASD patient brains ($p = 1.5e^{-26}$, Fisher's exact test, Fig. 4a, Supplementary Data 9). These include changes in GREs linked to genes previously proposed to play a role in ASD such as *c-MET*[46], *CITED4*[47], *NLK*[47], *CAMK2A*[47], and *DNMT3A*[47,48]. For example, two hominin-specific PFC gains that were reduced in ASD brains were linked to *DNMT3A* and *CAMK2A* by HiC data (Fig. 4d, c, Supplementary Fig. 8c, d), the latter of which was shown to regulate dendritic morphology and synaptic transmission[49] and is found mutated in autism patients[50]. In addition, two oligodendrocyte-specific GREs emerged in the c-MET locus (Fig. 4d, Supplementary Fig. 8e), a region proposed to play a role in ASD[46], with both of these elements showing a significant reduction of H3K27ac enrichment in ASD patients[44]. Enrichment

of H3K27ac at these GREs positively correlated with both *c-MET* expression levels and H3K27ac enrichment on the promoter (Fig. 4e, f)[44] and both GREs contact the *c-MET* promoter when assessed in 4C experiments using white matter tissue (Fig. 4g).

To further explore the link between hominin-specific gains and ASD losses, we analyzed whether ASD losses could be assigned to a specific cell type. Analyzing human NeuN− and NeuN+-specific open chromatin regions, we observed a slight reduction in glia-specific open chromatin and a gain in neuron-specific signal in PFC samples from patients compared to controls (Supplementary Fig. 9a). In contrast, AD brains showed a loss of NeuN+ signal that is consistent with a loss in neurons in the entorhinal cortex[51]. This was also observed using H3K27ac FANS data (Supplementary Fig. 9b). While the reduction in glia-specific signal in ASD patients could underlie a shift in cellular composition, they could also be related to cell-type-specific transcriptional programs that are changed. This is more likely as consistent changes in glial cell content in the PFC of adult autism patients were not observed previously[52–54]. Independent data from single-cell ATAC-Seq data confirmed enrichment for oligodendrocyte-specific open chromatin at regions that were lost in ASD (Fig. 4g). Nevertheless, regions that are specifically gained in hominins are substantially more often oligodendrocyte-specific compared to the regions lost in autism brains (Fig. 4g, Supplementary Fig. 9b). Moreover, LLM3D analysis[55] suggests that, while a part of the interaction between ASD and hominin changes may be driven by both of these preferentially affecting oligodendrocytes or a subtype thereof, the specificity of these regions is higher than expected based on a random interaction between the two sets ($p < 10e^{-16}$, LLM3D). Thus oligodendrocyte overrepresentation in the two sets is not the sole explanation for the interaction between hominin gains and ASD.

To analyze whether the link between evolution of novel GREs and ASD was specific to hominin-specific gains, we repeated the analysis for human and chimpanzee-specific regulatory changes. We defined these as consistent gains or losses of activity compared to all three other primate species (Supplementary Fig. 9c). We found that both human and chimpanzee-specific gains were also enriched for GREs depleted in ASD ($p = 4.0e^{-3}$ and $p = 1.2e^{-4}$ respectively, Fisher's exact test, Supplementary Fig. 9d). Cell-type analysis of these regions showed that both preferentially originate from oligodendrocyte (Supplementary Fig. 9e). Nevertheless, we observed a reduction in oligodendrocyte specificity for chimpanzee-specific regions that are lost in ASD compared to human-specific GREs, the latter being similar to hominin-specific gains (Supplementary Fig. 9e, f). Thus, the skewing of regions that evolved in hominins towards oligodendrocytes and ASD persisted during human but not chimpanzee evolution. Therefore, we propose that a regulatory program affecting oligodendrocytes past the postnatal stage, which is preferentially disrupted in ADS, first emerged in hominins and continued to change in the human lineage.

## Discussion

A connection between human evolution and the emergence of neural disease has long been suspected[4], with several genetic elements that recently evolved in humans or in great apes showing evidence for disease-related changes[2,3,5,56]. For instance, several human accelerated regions (HARs) were linked to both ASD and schizophrenia supporting a role for human-specific regulatory changes near neural genes in these psychiatric disorders[2,56]. As an increased susceptibility to neural disease is unlikely to have evolved in isolation without an added benefit, connecting human evolution to neural disorders may lead to unraveling of the key genetic changes that underlie the emergence

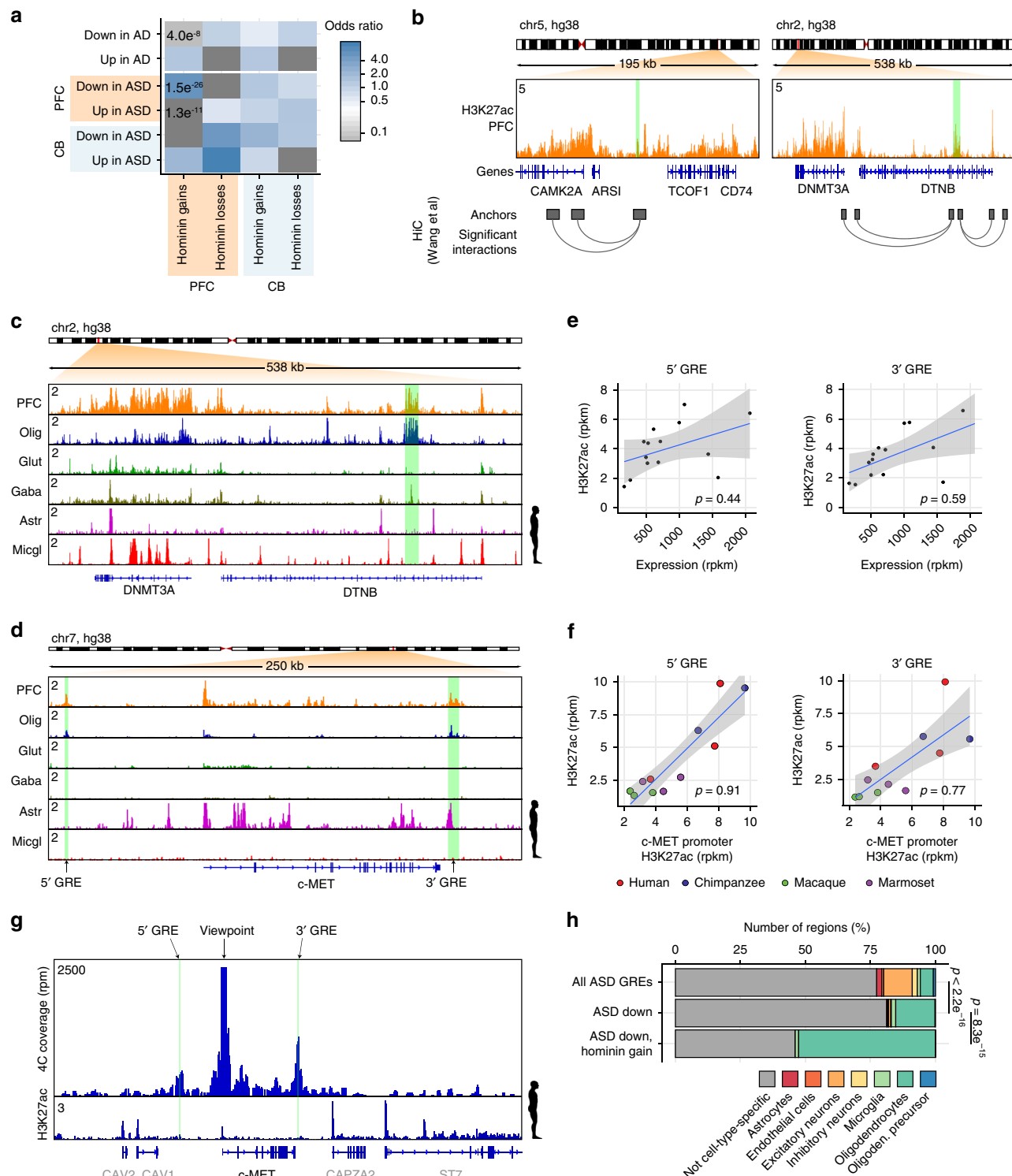

of the human brain. However, data on great apes are rare and as such human evolution is often inferred based on comparison of human and rhesus macaque[1]. Furthermore, it is far from clear what the defining features are that make the human brain special and to what extent these may or may not be shared with larger primates[8]. Our data demonstrate that the regulatory changes that set the stage for the human condition as well as some of their associated disorders may have emerged earlier, prior to the separation of human and chimpanzee. These changes correlate with hominin-specific sequence changes and selectively affect GREs in oligodendrocytes, key regulators of synaptic plasticity

throughout life and essential for higher executive function[57]. Furthermore, these regulatory elements are specifically employed between postnatal life and adulthood. Indeed, protracted myelination during postnatal development has been proposed as one of the hallmarks of human brain development compared to other primates[58]. Moreover, expression analysis suggests that regulatory programs affecting myelination recently changed in the human lineage[15]. Our data suggest that, while oligodendrocytes were selectively altered during hominin evolution, changes in oligodendrocytes continued to occur in the human lineage in elements that are relevant in ASD. This suggests that changes in this

**Fig. 4 Hominin-specific gains are selectively deregulated in autism patient brains. a** Heatmap showing enrichment of disease-associated GREs deregulated in ASD and AD patient brains correlating with hominin-specific changes found in cerebellum (CB) and prefrontal cortex (PFC). Color indicates odds ratio of enrichment compared to all identified GREs. *P* values were determined using a Fisher's exact test. **b** Chip-seq track showing rpm normalized H3K27ac enrichment across regions containing the *CAMK2A* (left panel) and *DNMT3A* (right panel) gene in human PFC. Hominin-specific PFC gains are highlighted in green. Chromosomal interactions are indicated by the boxes below the panel. **c** ChIP-seq tracks as in (**a**), for the *DNMT3A* gene in human PFC and different isolated cell types. A hominin-specific PFC gain is highlighted in green. PFC prefrontal cortex, Olig oligodendrocytes, Glut glutamatergic neurons, Gaba GABAergic neurons, Astr astrocytes, Micgl microglia. **d** ChIP-seq tracks as in (**b**) for a region containing the *c-MET* gene in human PFC and different isolated cell types. Two hominin-specific PFC gains are highlighted in green. **e** Correlation of H3K27ac enrichment in the 5′ and 3′ GRE with expression of the *c-MET* gene in autism patients. Black dots represent samples, blue line represents linear regression through the samples, gray area highlights the 95% confidence interval. **f** As in (**e**), correlation of H3K27ac enrichment on the 5′ and 3′ GRE with H3K27ac enrichment on the promoter. Dots represent the different samples, colors indicate species. **g** 4C-seq and rpkm normalized H3K27ac ChIP-seq tracks of human white matter in a region containing the *c-MET* gene. The *c-MET*-promoter was used as viewpoint for the 4C analysis. The green highlighted areas indicate the location of hominin-specific PFC gains as shown in (**d**). **h** Bar plot showing the cell-type specificity of regions based on single-cell ATAC-Seq data, on regions identified in ASD patients and control brains, regions specifically downregulated in ASD patients and those that are down in ASD as well as hominin-specific gain in PFC. Difference in oligodendrocyte-specific frequency was calculated using a Fisher's exact test. Source data are provided in Source Data file.

regulatory program may have been important both during the evolution of hominins as well as the emergence of humans.

Defects in oligodendrocyte function are gaining attention as a contributing factor to a variety of neurodegenerative and neuropsychiatric diseases[57]. While the evolutionary changes in oligodendrocytes were linked to hominin-specific sequence changes, and not due to changes in oligodendrocyte content, it is unlikely that parallel mutations in sets of recently evolved regulatory elements in oligodendrocytes occur at the same time in ASD patients. Thus, the link between hominin evolution and ASD presented here may reflect a recently evolved oligodendrocyte-specific transcriptional program that is driven by the regulatory elements identified and that is preferentially affected in ASD or a specific oligodendrocyte subtype[59]. As such, analysis of oligodendrocyte subpopulations as well as regulatory changes linked to ASD risk genes such as *CAMK2A* and *DNMT3A*[47,48] may further our understanding of oligodendrocyte function in ASD. In addition, *c-MET*[46], a gene previously proposed to play a role in ASD, is of interest as it codes for a receptor tyrosine kinase that is targeted by hepatocyte growth factor (HGF), which can enhance the proliferation and migration of oligodendrocyte progenitors and delay their differentiation[60]. A single nucleotide variation in the *c-MET* promoter reduces *c-MET* expression and protein levels and was associated with an increased ASD risk and altered connectivity in ASD patient brains[61,62]. Furthermore c-MET protein levels were found reduced in autism brain samples[63]. Interestingly, both HGF expression as well as *c-MET* expression were shown to also increase the migration and proliferation of oligodendrocyte progenitor cells in tissue culture[60] and expression of HGF by oligodendrocyte precursors was shown to enhance neural survival[64]. In line with these observations, neural growth and dendrite and synapse maturation is dependent on MET receptor signaling[65]. Nonetheless, levels of *c-MET* expression were not significantly altered in a recent study analyzing the PFC of autism patients[66]. Thus the potential role of this gene and its control of oligodendrocyte differentiation in ASD requires further study. As such our data provide important insight into the regulatory changes that have contributed to the emergence of the human brain, while prioritizing a new set of regulatory elements that could underlie some of the pathology observed in ASD patients.

## Methods

**Sample and data collection**. Marmoset (*Callithrix jacchus*, cj1, cj2, cj3) tissue samples were collected at the Biomedical Primate Research Centre (BPRC) in Rijswijk, the Netherlands (http://www.bprc.nl) and represent rest material involving no animal experimentation for the purpose of this work as determined by the Animal Experimental Committee (DEC) (Supplementary Data 1). Samples were flash frozen immediately after dissection and stored at −80 °C. Three brain regions, cerebellum, prefrontal cortex and white matter, were used for analysis in this

manuscript. Brain samples from human (*Homo sapiens*; hg1, hg2, hg3), chimpanzee (*Pan troglodytes*; pt1, pt2) and rhesus macaque (*Macaca mulatta*; rm1, rm2, rm3) have been generated by us previously[14,67], or were resampled from the original brains. The adult nondemented control female human hemispheres were obtained from the Netherlands Brain Bank (http://www.brainbank.nl/). Informed consent was acquired meeting all ethical and legal requirements for autopsy, tissue storage, and use of tissue and clinical data for research. Other datasets analyzed in this study were obtained from GEO (https://www.ncbi.nlm.nih.gov/geo/) or Synapse (https://www.synapse.org) (Supplementary Data 2).

**Chromatin immunoprecipitation followed by sequencing**. Chromatin immunoprecipitation (ChIP) was performed as previously described by us[14,67]. Sixty milligrams of tissue was used per ChIP and homogenized in a glass douncer (Kontes Glass Co.) in 1 ml Dulbecco's Modified Eagle Medium with 0.2% Bovine Serum Albumine (BSA). Cells were crosslinked while rotating for 10 min at RT in 10 ml fixation buffer (freshly made; 1% formaldehyde, 0.5 mM ethylenediaminetetraacetic acid (EDTA), 0.05 mM ethylene glycol-bis(β-aminoethyl ether)-N,N,N′, N′-tetraacetic acid (EGTA), 10 mM NaCl, 5 mM HEPES-KOH, pH 7.5). Samples were washed twice with PBS, centrifuged for 5 min at 2095 × g and 4 °C, resuspended in 10 ml lysis buffer (50 mM HEPES-KOH pH 7.5, 140 mM NaCl, 1 mM EDTA, 10% glycerol, 0.5% Igepal, 0.25% Triton X-100) and incubated for 10 min at RT while rotating. Cells were then centrifuged for 5 min at 2095 × g at 4 °C and resuspended in 10 ml wash buffer (200 mM NaCl, 1 mM EDTA, 0.5 mM EGTA, 10 mM Tris-HCl pH 8.0) and incubated for 10 min at RT while rotating. Cells were pelleted for 5 min at 2095 × g at 4 °C and resuspended in 150 μl sonication buffer (1 mM EDTA, 0.5 mM EGTA, 10 mM Tris-HCl pH 8.0, 100 mM NaCl, 0.1% Na-Deoxycholate, 0.5% *N*-lauroyl sarcosine), divided over two microtubes (Covaris 520045) and sonicated using the Covaris S series (12 cycles of 60 s: intensity 3, duty cycle 20%, 200 cycles/burst). After sonication, the multiple microtubes per sample were pooled and sonication buffer and Triton X-100 (final concentration 1%) was added to a total volume of 550 μl. Immunoprecipitation with antibody (H3K27ac: ab4729 abcam, H3K4me3: ab8580 abcam) coated Dynabeads Protein G (Invitrogen 10003D) was performed overnight at 4 °C. Beads were subsequently washed four times with RIPA (50 mM HEPES-KOH pH 7.5, 1 mM EDTA, 0.7% DOC, 1% NP40 and 0.5 M LiCl) and once with 50 mM NaCl in TE. Elution of the DNA from the beads was performed overnight in elution buffer (50 mM Tris pH 8.0, 10 mM EDTA, 1% Sodium Dodecyl Sulfate (SDS)) at 65 °C. Beads were removed by short centrifugation and the supernatant 1:1 diluted with TE, followed by a 2 h incubation with RNase (final concentration 0.2 μg/μl) at 37 °C and a 2 h incubation with proteinase K (final concentration 0.2 μg/μl) at 55 °C. Finally, the DNA was extracted using phenol/chloroform with MaXtract High Density gel tubes (Qiagen) and purified using ethanol. Sequencing libraries were prepared according to the Illumina Truseq DNA library protocol and samples were sequenced at the MIT BioMicro Center (http://openwetware.org/wiki/BioMicroCenter) using the Illumina HiSeq 2000 sequencer.

**RNA sequencing**. White matter tissue for bulk RNA sequencing was dissected from each primate species (100 mg per sample). Tissue was cut into small pieces of 1 mm length and collected in 1 ml Trizol. Samples were vortexed to dissolve the tissue followed by a 5-min incubation at RT. Samples were centrifuged for 5 min at 21,000 × g at 4 °C and the supernatant was transferred to a fresh Eppendorf tube. Four hundred microliters chloroform was added and samples were mixed by shaking, incubated on ice for 5 min and centrifuged for 15 min at 21,000 × g at 4 °C. The aqueous upper layer was transferred to a fresh tube. Five hundred microliters iso-propanol and 1 μl glycoblue (Invitrogen) were added followed by mixing by shaking and incubation overnight at −20 °C. The following day, samples were centrifuged for 30 min at 21,000 × g, 4 °C. RNA pellets were washed twice with 75% ethanol and air-dried for 8 min at RT. Pellets were resuspended in 11 μl

nuclease-free water and transferred to USEQ (www.useq.nl) for library preparation and sequencing. RNA libraries were prepared according to the Illumina TruSeq Stranded total RNA library protocol and samples were sequenced at USEQ on a high-output NextSeq500, 1 × 75 bp.

**ATAC-sequencing on FANS sorted nuclei**. Prefrontal cortex tissue was dissected from the primate brains and homogenized in a glass douncer (Kontes Glass Co.) in 2 ml EZ buffer (Nuclei Isolation Kit, Sigma NUC101) and incubated for 5 min on ice. Subsequently samples were centrifuged for 15 min at $65 \times g$ at 4 °C, resuspended in 2 ml EZ buffer and incubated on ice for 5 min. Samples were again centrifuged for 10 min at $65 \times g$ at 4 °C, resuspended in 4 ml Nuclear suspension buffer (NSB; 0.01% BSA, 1× complete protease inhibitor cocktail in PBS) and filtered through a 40 µm cell strainer. Samples were centrifuged for 15 min at $65 \times g$ at 4 °C and resuspended in 30 ml NSB after which 10 ml 30% OptiPrep (Simga D1556) was added on the bottom of the sample using a needle. Another layer of 5 ml 60% OptiPrep was then added underneath and the sample was centrifuged for 10 min at $1000 \times g$ at 4 °C. Purified nuclei were recovered from between the 60% OptiPrep and PBS layers and checked for quality under a bright field microscope. One hundred microliters blocking solution (0.5% BSA, 10% FBS in PBS) was added containing anti-NeuN antibody conjugated with Alexa488 (1:1000, Merck Millipore MAB377X) and Hoechst 34580 (1 µg/ml, Fisher Scienctific H21486) and incubated for 1 h at 4 °C in the dark on a roller. Stained nuclei were then transferred to FACS tubes precoated with 5% BSA and sorted on NeuN signal using a FACS-Jazz (BD Bioscience). 50,000 NeuN+ and NeuN− stained nuclei were collected in PBS and processed further for ATAC-sequencing[68]. In short, 0.1% NP-40 was added to the sorted nuclei and samples were centrifuged for 15 min at $150 \times g$ at 4 °C. Pellets were then resuspended in 1 ml resuspension buffer (10 mM Tris-HCl pH 7.5, 10 mM NaCl, 3 mM MgCl$_2$, 0.1% Tween-20, 0.1% NP-40 in MQ) and incubated for 3 min on ice. One milliliter resuspension buffer without NP-40 was then added to the sample and centrifuged for 10 min at $500 \times g$ at 4 °C. Nuclei were resuspended in 50 µl transposition mix (1x TD buffer (20 mM Tris-HCl pH 7.6, 10 mM MgCl$_2$, 20% dimethyl formamide in MQ), 100 nM Tn5 transposase, 33% PBS, 0.01% digitonin, 0.1% Tween-20 in MQ) and incubated for 30 min at 37 °C while shaking at 1000 rotations per minute. Samples were purified using Qiagen MinElute PCR purification kit (Qiagen 28004) and eluted in 21 µl MQ. Purified DNA was amplified with NEBnext High-Fidelity PCR master mix (NEB M0541S) and appropriate sequencing adapters for five PCR cycles. Library complexity was determined by qPCR on 5 µl of the PCR sample and the number of extra PCR cycles determined[69]. PCR samples were purified using AMPure XP beads (Agencount A63881), eluted in 12 µl MQ and sequenced at USEQ on a high-output NextSeq500, 1× 75 bp.

**Mapping and analysis of primate ChIP-Seq data**. To ensure proper comparison of samples from different platforms and studies, all reads from primate cortical and cerebellar samples were trimmed to a 36 bp length using the Fastx-toolkit (http://hannonlab.cshl.edu/fastx_toolkit/index.html). Sequences were aligned using Bowtie 1.1.2 (http://bowtie-bio.sourceforge.net/index.shtml) excluding reads with more than one mismatch (seed length 36) or with multiple alignments, unless stated otherwise. Reads were mapped to the most recent reference genome available (Supplementary Data 3). Fraction of reads in peaks (FRiP) scores all exceed the 1% threshold used by the Encyclopedia of DNA Elements (ENCODE)[21], and the percentage of unique reads was overall high (mean 89.25%, Supplementary Data 3). Genome-wide enriched regions were annotated per sample using MACS2 version 2.1.1 ($p = 10^{-5}$, extsize = 300, local lambda = 100,000). Whole-cell extract input controls were generated for each common marmoset brain region (Supplementary Data 3). We used the internal MACS2 lambda control to correct for local bias as whole-cell extract inputs often introduce sonication biases at open chromatin regions. Identified H3K27ac or H3K4me3 enriched regions were extended to a minimum size of 2000 bp (peak center ± 1000 bp), the resolution typically observed for these regions[13,67]. Lists of enriched regions per species and brain region were obtained by merging the identified regions of the replicates per tissue, with regions overlapping at least 1 bp being stitched together. This lenient cutoff was chosen to ensure that most enriched regions with multiple summits were merged. The average overlap was ~700 bp, with only 3% of regions were a merger with an overlap of less than 100 bp. Reproducible enriched regions were defined as enriched regions present in at least two biological replicates of the same brain region per species. Nonredundant H3K27ac-enriched region lists were obtained per primate species by merging the regions of both brain regions. Public data were downloaded from either the GEO or Synapse repositories (Supplementary Data 2) and reanalyzed to match our analysis using the most recent primate genome if required, using the settings described above.

**GRE analysis and quality control across primate genomes**. Primate regulatory elements were defined using the UCSC liftOver tool (−minMatch = 0.1) (Supplementary Fig. 2c). H3K27ac-enriched regions of chimpanzee, rhesus macaque and marmosets were reciprocally mapped to the human genome (hg38). Regions that were mapped to multiple (nonunique) locations and regions that changed more than 50% in size were excluded ($n = 2043$). Following mapping to hg38, all four lists for the different primate species were merged. Average overlap was

~2700 bp, with only 0.3% of regions were a merger with an overlap of less than 100 bp. The resulting list was again mapped to all three primate genomes using reciprocal liftover to make sure that merged regions were mappable across all primates. To ensure equal mappability, >90% of the bases within a regulatory element had to be properly annotated in all reference genomes (<10% overlap with UCSC Table Brower's gap lists). This cutoff was chosen as unknown bases generally occur in stretches rather than as single nucleotides across the genome. To account for repetitive or duplicated genomic regions, which are especially susceptible to poor annotation in lower-quality genomes, enrichment scores were not allowed to change significantly in the target genome when allowing reads to map to multiple locations. These repetitive regions were defined per species by mapping reads from every sample to unique locations (bowtie: –best –strata –m 1) as well as to multiple locations (bowtie: –best –strata –M 1). Genomic regions that were enriched using the multimap settings but not with unique mapping are potential repetitive elements that are not annotated at similar depth across all the genomes. All regulatory elements that overlap a repetitive element on any of the primate species ($n = 11,432$) were therefore discarded. In total, 37,308 regulatory elements could be identified with the above restrictions on all four genomes (Supplementary Data 4). GREs were assigned to their target gene based on their location on the human genome as it is the best annotated genome. GREs located within 1000 bp of an annotated TSS (USCS RefSeq hg38) were considered promoters and assigned to the associated gene. Enriched regions located outside 1000 bp from a TSS were classified as putative enhancers and were assigned to a target gene based on PFC HiC data[9]. If no significant enhancer−promoter loop was annotated, the enhancer was assigned to its closest active TSS based on H3K4me3 enrichment (Supplementary Data 5).

**Hierarchical clustering, PCA, t-SNE analysis**. Duplicate reads were removed from the bam files using Samtools 1.3.1 and read coverage within enriched regions was counted using Bedtools v2.26.0. Read counts were then normalized for the total number of uniquely mapped reads per sample and log2 transformed using the rlog (blind) function in DESeq2. For hierarchical clustering (e.g. Supplementary Fig. 2b, f), Pearson correlations between the samples were calculated. Samples were clustered based on Pearson distance using average linkage and heatmaps were generated using the heatmap.2 function from the gplots R package. For principle component analysis (PCA, Fig. 1c), the prcomp function in R (http://www.R-project.org) was used. t-SNE multidimensional scaling coordinates were determined using the t-SNE R package (Fig. 1b). H3K27ac heatmaps and metaplots were generated using the ngs.plot.r (e.g. Supplementary Fig. 1c).

**Annotation of hominin-specific regulatory changes**. Hominin-specific regulatory changes were identified based on differential H3K27ac-enrichment between great ape (human and chimpanzee) and monkey (rhesus and marmoset) species. Pairwise comparison of human and chimpanzee with macaque and marmoset for both cerebellum and prefrontal cortex was performed using DESeq2 (Supplementary Fig. 3a, b). Per pairwise comparison we temporarily excluded all regions with zero original read count in all the replicates of a single species; in total, 711 regions were excluded. Raw read counts were then used to define differentially enriched regions using the DESeq2 function in R. Regulatory elements with a twofold change in enrichment and an FDR < 0.01 were defined as significantly differentially enriched (DE). Hominin regulatory changes were defined as all regions that are DE for both human and chimpanzee compared to macaque and marmoset, or vice versa (Supplementary Fig. 4a, Supplementary Data 6).

**Analysis of GRE cell-type specificity**. To assess the cell-type specificity of GREs in our datasets, we leveraged ChIP-seq data from FANS sorted PFC nuclei[27,30] and single-cell ATAC-seq data from PFC[34]. For FANS sorted nuclei, paired-end ChIP-seq sequencing reads were obtained and mapped to the human genome using Bowtie 1.1.2, excluding reads with more than one mismatch, multiple alignments or where one of the pairs could not be mapped. Metaplots and heatmaps were obtained using the ngs.plot.r function.

For cell-type-specific ATAC-Seq regions, we obtained lists of differentially enriched regions per cell type as provided[34]. From these we selected regions that were differentially enriched in a single-cell type. GREs from out datasets were then overlapped with these cell-type-specific ATAC-Seq regions and GREs overlapping a single-cell-type-specific ATAC-Seq region were annotated as specific to that cell type. GREs overlapped none or more than one cell-type-specific ATAC-Seq region were annotated as not cell-type-specific. The difference in oligodendrocyte-specific content between sets of GREs was calculated using a Fisher's exact test in R.

**Confounder analysis and batch effect**. Batch effect was controlled for as done previously by analyzing biological replicates in separate batches[14,67]. This reduces consistency in batch-related influences between replicates thus reducing the chance that these are picked up in DE analysis. This also allows for the reanalysis of data that have been generated at an earlier stage and have been handled similarly. To assess the effect of confounding variables, such as post-mortem delay (PMD) and sequencing batch between our datasets, we performed multivariate analysis on the H3K27ac enrichment of the here identified GREs. We defined several biological and technical covariates that could have influenced enrichment scores, including

species, tissue type, gender, PMD, sequencing depth, sequencing platform, FRIP score and experimental batch. These covariates were used in a linear regression model on rpkm normalized read tables, after which we used ANOVA to extract the percentage of variance explained and significance per covariate for every GRE (Supplementary Fig. 3c−e). Only 27 out of 1930 GREs that changed in hominins (gains and losses) had a significant contribution of covariates (batch, gender, PMD, sequencing platform, $p < 0.01$, Supplementary Fig. 3d). One of these was also deregulated in ASD brain. To further exclude that the identified characteristics of hominin-specific PFC gains are not due to batch effect, we analyzed white matter ChIP-sequencing samples where the samples were equally distributed across pre-defined experimental batches with all of the different species occupying the same batch and batches containing biological replicates. The resulting regions were still enriched for oligodendrocytes and preferentially deregulated in ASD brain (Supplementary Fig. 9g, h), excluding batch as a basis for our observations.

**Functional analysis of gene sets and motif analysis**. Gene ontology analysis was done using the Genomic Regions Enrichment of Annotations Tool (GREAT version 3.0.0, http://bejerano.stanford.edu/great/public/html) with basal plus extension setting. Multiple genes can therefore be assigned to the supplied enriched regions (Fig. 3d, Supplementary Fig. 6f, Supplementary Data 8). Motif enrichment was performed on hominin-specific regulatory changes for cerebellum and prefrontal cortex separately, using HOMER version 4.9.1 with default parameters (Supplementary Data 7).

**Stratified LD score regression of SNPs in hominin GREs**. To study enrichment for disease heritability at hominin-specific GREs, we obtained the summary statistics of genome-wide associate data for various brain-related phenotypes (Alzheimer's disease, ASD, Parkinson's disease, schizophrenia and bipolar disorder). To calculate disease enrichment, we added our sets of GREs (hominin-specific changes, all GREs, all enhancers, all promoters) and ASD-associated GREs as defined previously[44], to the baseline model of this package. We also included a set of randomly shuffled GREs. Enrichment statistics were calculated as the ratio between explained heritability and the proportion of SNPs in H3K27ac-enriched regions, using the LD score regression software (https://github.com/bulik/ldsc).

Autism spectrum disorder[44] or Alzheimer's disease[45]-associated regulatory elements were categorized as more enriched or less enriched in patients brains over control samples using recent data. The resulting regions were compared with hominin gains and losses from cerebellum and prefrontal cortex. Significant enrichment or depletion compared to all annotated GREs per brain region was determined using a Fisher's exact test in R. The statistical significance cutoff was set at 0.01 and P values were adjusted for multiple testing using the Benjamini −Hochber's FDR method.

**Sequence conservation and hominin nucleotide changes**. PhastCon scores were calculated using the UCSC phastCon 20 mammals track (http://hgdownload.soe.ucsc.edu/goldenPath/hg38/phastCons20way/). To increase the resolution at GREs for relevant nucleotides, we used only those nucleotides that were DNAseI enriched in the frontal cortex, as defined by ENCODE regions within the GREs, to calculate conservation[29]. Mean values of all scored nucleotides per GRE are plotted (Fig. 1e). Hominin-specific nucleotide changes were defined based on the pairwise alignment files of chimpanzee, rhesus macaque and marmoset with human, as obtained from UCSC. All nucleotides that shared the same base between human and chimpanzee but not with macaque and marmoset were annotated as hominin-specific nucleotide changes. The percentage of hominin-specific nucleotide changes per GRE is plotted (Fig. 1f).

**Analysis of gene expression data and open chromatin**. Raw RNA sequencing reads were trimmed to a length of 60 bp, starting from base 12, using the Fastx-toolkit and aligned to the appropriate genome using hisat2 2.0.5 (https://ccb.jhu.edu/software/hisat2/index.shtml) using default settings (Supplementary Data 3). Duplicated reads that are likely PCR duplicates were removed. Reads mapping to multiple locations were also removed from further analysis. Expression values were counted for each gene using the featureCount function of the Rsubread packages in R. ATAC-sequencing reads were trimmed to remove Nextera adapter sequence 5′-CTGTCTCTTATA-3′ using cutadapt (https://cutadapt.readthedocs.io/en/stable/) and processed similar to ChIP data as described above.

**Analysis of interdependence**. The interdependence between categorical variables (hominin-specific gain, deregulated in ASD and oligodendrocyte specificity) of the identified PFC GREs was calculated using LLM3D[55]. LLM3D fits a number of log-linear models to 3D contingency tables of GRE counts and selects the model that best fits the observed element characteristics. These models imply different (in) dependence relationships between the variables, with the null hypothesis assuming complete independence. The significant P value indicates that the oligodendrocyte specificity of the hominin-specific gains deregulated in ASD is higher than expected for a random overlap between the two sets.

**Chromosome Conformation Capture (4C) analysis**. Frozen white matter tissue was dissected from the brain and pulverized in liquid nitrogen using a pestle and mortar. The frozen tissue was further homogenized in 1 ml PBS with 10% fetal bovine serum (FBS) using a cold 2 ml dounce (Kontes Glass Co.). Cells were then crosslinked and processed, using DpnII and Csp6I as restriction enzymes. Cells were fixed for 10 min on RT in fixation buffer (10% FBS, 2% formaldehyde in PBS), after which glycine was added to a final concentration of 125 mM to quench the fixation. Samples were washed twice with PBS and centrifuged for 5 min at $750 \times g$. Cells were lysed in 2 ml cold lyses buffer (50 mM Tris-HCl pH 7.5, 150 mM NaCl, 5 mM EDTA, 0.5% NP-40, 1% Triton X-100) and incubated for 5 min at RT, 5 min at 65 °C and 1 min on ice. Samples were subsequently pelleted by centrifugation for 5 min at $550 \times g$. The 3C template was obtained by addition of restriction buffer (1× DpnII restriction buffer, 3% SDS in MQ) and incubated for 1 h at 37 °C. Triton X-100 was added to a 2.5% concentration and samples were again incubated for 1 h at 37 °C. Restriction of the samples was performed overnight at 37 °C after addition of 10U DpnII restriction enzyme, followed by an overnight ligation at 15 °C after addition of T4 ligation buffer (final concentration: 1×) and 1.5 μl T4 ligase. The following day, 100 μg proteinase K was added and samples incubated overnight at 65 °C, followed by a 45 min incubation at 37 °C with 100 μg RNase A. The 3C template was purified using phenol-chloroform and processed for another round of restriction-ligation using Csp6I to create the 4C template. For amplification on the MET promoter viewpoint, the following primers were used: 5′- TACACGACGC TCTTCCGATCTCTAATGAATTTTTTCTGCATGAAGAT-3′ as reading primer and 5′- ACTGGAGTTCAGACGTGTGCTCTTCCGATCTGTTACCAGCCCTAG ACGTG-3′ as nonreading primer. Sequencing was done on the Illumina MiniSeq platform. Sequencing reads were trimmed to remove primer sequences and mapped on hg19, removing all reads that mapped to multiple locations. 4C coverage was calculated by averaging mapped reads in running windows of 41 fragments ends.

**Reporting summary**. Further information on research design is available in the Nature Research Reporting Summary linked to this article.

## Data availability
The ChIP-seq and RNA-Seq data reported in this study are available at Gene Expression Omnibus with accession code GSE130871. Other public datasets sets employed in our study can be found via accession numbers in Supplementary Data 2.

## Code availability
All analysis was done using R (http://www.r-project.org) or Python (http://www.python.org) employing the packages described above.

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

## Acknowledgements

M.P.C. was supported by the Royal Academy of Sciences (KNAW) in the Netherlands and the Erasmus Medical Center. We thank Utrecht Sequencing Facility for providing sequencing service and data. Utrecht Sequencing Facility is subsidized by the University Medical Center Utrecht, Hubrecht Institute, Utrecht University and The Netherlands X-omics Initiative (NWO). We thank Kevin Kenna and Mark Bakker from UMC Utrecht Neurogenetics department for their help with the stratified LD score regression analysis.

## Author contributions

B.C. and M.P.C. conceived and designed the experiments. B.C., I.S.T., C.R.M.W., M.W.V. and P.S. performed the experiments and were supervised by M.P.C. and N.G. I.K. collected and dissected primate brain specimens. B.C., M.L.B., G.G. and V.B. analyzed the data and were supervised by M.P.C. and W.d.L. B.C. and M.P.C. wrote the manuscript.

## Competing interests

The authors declare no competing interests.

**Additional information**

