## [Peer Review File · Nature Communications]

Reviewers' Comments:

Reviewer #1:

Remarks to the Author:

This manuscript by Castelijn et al generates new marmoset ChIP-seq and RNA-seq datasets and incorporates these data with previously generated ones from the same group from humans, chimpanzees, and macaques (Vermunt et al). The authors use these data to identify hominid (shared between humans and chimpanzees) gene regulatory elements (GREs). They analyzed prefrontal cortex and cerebellum as previously done and investigated GRE evolution using ChIP-seq for histone H3 lysine 27 acetylation. The authors also analyze H3K27ac data derived from FACS sorted nuclei from different cell types as well as publicly available H3K27ac from autism patients and healthy controls.

The authors find that the hominid GREs are notable for two reasons: 1) the gained ones are enriched in oligodendrocytes and 2) the gained ones are also overrepresented among regions that lose H3K27ac enrichment in ASD brains.

The authors also note that hominid gains are associated with regulatory regions that gain H3K27ac marks in postnatal and adult tissues compared with prenatal tissues, which further tracks with the idea that these marks are important for glia and not neuronal development which happens mostly prenatally.

In general, these results are important for our overall understanding of GRE evolution and the relative contribution of different cell types to this process. However, there are some concerns that need to be addressed.

Major comments:

1) Given that the authors focus on hominid alterations, the rationale for comparison to disease-relevant data is a bit confusing. While there is certainly no way in which we could determine whether chimpanzees develop ASD, and chimps do develop pathological hallmarks of other brain disorders such as Alzheimer's disease, it seems potentially a comparison that might result in false positives. My preference would have been for the authors to define human-specific GREs and compare those to the disease-relevant datasets. In the abstract the authors make references to speciation and the human brain, so it is a bit misleading for the paper to then focus on hominid alterations. I therefore suggest modifying the abstract unless the human-specific changes will indeed be explored.

2) The hominid-specific regions (1398 gained and 532 lost) are the sum of PFC and CB hominid-specific regions. This is not really mentioned in the main text where it should be. There is a discontinuity between the main text and the figures and this might be confusing for the reader. I would be more specific regarding PFC/CB and provide specific numbers when necessary. It appears the finding related to oligodendrocytes is only relevant to data derived from the cortex and not the cerebellum? If this is the case, this should be more explicitly discussed.

3) It is surprising that there is no postnatal-adult enrichment of All GRE regions given that these were defined from adult individuals (Fig 2d). Can the authors comment on this a bit more? Could this be related to how these regions are identified? Maybe the regional interval ~5kb is too broad?

4) Overlapping regions: what is the number of nucleotides that overlap to declare a peak overlapping?

5) Are biological/technical covariates taken into account?

6) What is the rationale for removing the duplicates from the RNA-seq data?

7) The use of the NeuN- and white matter data to support the idea that the results are not simply due to alterations in glia/neuron ratios is not well explained. The authors should detail the logic behind these comparisons more clearly.

8) The concept of taking overlaps of pairwise comparisons for the definition of hominid-specific changes does not take into account evolutionary distances of the species/parsimony. Please address this.

Minor comments:

1) The heatmap in figure 1d might be improved if the authors scaled the gain and loss heatmaps the same way so that the difference in the numbers of gained and lost regions across tissues was more intuitive. Currently, it looks like there are equal numbers if you are only looking at the heatmaps.

2) Extended figure 2f suggests that cerebellar GREs might be more divergent between hominids and monkeys than PFC GREs. Was there anything interesting from the cerebellum specific GREs to discuss?

3) How do the authors interpret the under-enrichment of gained GREs among regions with more H3K27ac signal in ASD patients in figure 3a?

4) The authors write that they removed regions that had zero log normalized read counts in one species. This is done prior to differential enrichment. How many GREs are removed at this step? It should also be clarified that hominid specific DE regions are compared to all GREs after this filtering step.

5) The manuscript should be proofread as it contains many typos. Also selection of words can be improved in some sentences (e.g. "existing DNA").

6) Labels of the x-axis are wrong in Extended Data Fig3b. It should be chimpanzee not human.

7) Figure 2a: there is a typo in the y-axis. "Gians" should be "Gains"; "Astrocytes" in Ext. Data in Fig4b.

8) What is the sample size on Fig2d? In other words, how many reads from each developmental group map to hominid gains and all GRE?

Reviewer #2:

Remarks to the Author:

Castelijns et al. have compared gene regulatory elements (GREs) in monkeys versus great apes to identify hominid-specific GREs (or hominid-gains). The basic assumption is that hominid-gains will provide important information about human evolution and higher cognitive abilities of human. Hominid-gains display oligodendrocyte-specificity and are altered in ASD postmortem brains. In particular, the latter finding is in line with a series of recent papers showing that human specific evolutionary elements are linked with neuropsychiatric and neurodevelopmental disorders. However, two major claims of this paper (that they identified hominid gains and they are associated with ASD) are somewhat compromised (the datasets have potential batch effects and the link to ASD is weak), which will require significant refinement. Below I framed my concerns in more details:

Major concerns

1. The major concern about this paper is that they used their new dataset as well as publicly available dataset (Vermunt et al), while not corrected for batches. While authors have pointed out that the same protocol was used as Vermunt et al., batches are not only caused by the protocol, but the person who ran the experiment, reagents used for the study (e.g. whether they came from the same lot number or when they were open), the day that the sequencing was run, and many other confounders. Since both datasets were generated by the same group, is this dataset newly generated or generated at the same time as Vermunt et al.?

This type of study design may be fine if they compare peaks vs. no peak, but can be a serious problem when they run differential peak analyses (which they ran). This is a complicated issue, as batch totally corroborates with samples, and batch correction can dilute the biological signals. It is

important for authors to exclude the possibility of the main finding being caused by batch effects.

2. Authors have not mentioned about whether they regressed out any confounding variables while running DESeq2. Comprehensive codes they used to perform the analysis will be helpful (please also see comment #12).

3. Rather than putting PC1 and PC2 separately for Figure 1c, can authors put them in the same plot as Figure 1b? This will give a clearer clustering pattern among all groups.

4. How many peaks did they find that are not changing across evolution (detected for all four species = stable GRE)? A more comprehensive picture will arise if authors can provide # of peaks that fall onto every sector of the Venn diagram in Extended Data Fig. 3c.

5. Can authors repeat Figure 2a-b and Figure 2d for stable GREs? It would be informative to see how species specific elements differ compared to the ones that do not change across evolution. Also, do authors have explanation of why microglia are depleted of any GREs in Figure 2b? Is this because of the quality of H3K27ac datasets in microglia?

6. It is hard to compare Figure 2b results between hominid-gains and all GREs because the scales differ, but according to Extended Figure 4c, it seems that NeuN+ read counts are smaller for hominid-gains than all GREs. If this is really true, does it mean that hominid-gains are under-represented for neuronal peaks?

7. Currently the basic genomic composition of hominid gains/stable GREs is unclear. How many of them overlap with promoters and exons? How many of them are located in the non-coding region (intronic/intergenic)? If the majority of them are intergenic and intronic, given that H3K27ac marks enhancers that often regulate distal genes via long-range interactions, I recommend authors to identify putative target genes based chromatin interactions, and repeat the analysis in Figure 2e and Extended Data Figure 4g. Authors have already done this in Figure 3f for a MET locus. As chromatin contact profiles in the brain tissue (<https://www.ncbi.nlm.nih.gov/pubmed/30545857>) and brain cell types (<https://www.ncbi.nlm.nih.gov/pubmed/30545851>) now became available, they can perform a similar analysis in a genome-wide fashion. There's also a paper that linked regulatory elements reported by Vermunt et al. to their target genes using Hi-C data: <https://www.ncbi.nlm.nih.gov/pubmed/31160561>.

8. It seems that GOs enriched for hominid-specific changes in the PFC and Cerebellum mark widely different processes. I couldn't find Figure 1e-f and 2a-e equivalent for hominid-gains in the cerebellum. Do hominid-gains in cerebellum exhibit different cell-type specificity? What's authors explanation to this?

9. Extended Data Figure 5b is interesting as it seems that a hominid-gain is linked to the same gene in all four species. How often this is the case that hominid-gains are linked to the same gene in all species? Is there a case that they mark different genes in different species?

10. Another major concern is that the link between hominid-gains and ASD is weak.

10-1. First, there was no enrichment of ASD GWAS signals in hominid-gains. This can be partly because authors used lead SNPs from the NIH GRASP, which has a limited number of hits for ASD GWAS. Therefore, I strongly encourage authors to use stratified LD score regression (<https://www.ncbi.nlm.nih.gov/pubmed/26414678>) to correct for LD and take genome-wide signals into account.

10-2. Second, ASD H3K27ac data that authors used <https://www.ncbi.nlm.nih.gov/pubmed/27863250> often display opposite direction to what we

would expect from the expression data (e.g. neuronal genes were downregulated in ASD postmortem brains, while H3K27ac peaks were upregulated). Consequently, it is currently unclear whether ASD differential H3K27ac peaks mark causal or responsive processes. Moreover, ASD-upregulated H3K27ac peaks were enriched for neuronal signals, while ASD-downregulated H3K27ac peaks were largely enriched for glial signals. Therefore, there is a chance that the enrichment in Figure 3a just captures cell-type specificity of ASD H3K27ac signals (e.g. ASD-downregulated H3K27ac peaks largely mark oligodendrocyte signals, which was consequently captured via hominid-gains). Therefore, additional layer of evidence will be necessary to support this claim.

A few suggestions I have are the followings: Once authors identify the target genes of hominid-gains (as suggested in comment #7), they can check whether those genes are enriched for ASD risk genes. Authors can also repeat the process with other brain disorder-relevant H3K27ac signals (e.g. Alzheimer's disease; <https://www.ncbi.nlm.nih.gov/pubmed/30349106>), which will provide disease-specificity of the enrichment. I believe there will be also other ways to bolden the claim.

10-3. Third, while the level of convergence of functional genomic evidence at the MET locus is remarkable (4C experiment, GRE H3K27ac vs. MET gene expression, GRE H3K27ac vs. MET promoter H3K27ac), its link to ASD is not very promising. For example, I don't think MET expression differences between control and patients will survive multiple testing correction of a typical DEG analysis. Authors neither corrected for potential confounders such as RIN, batches, and PMI when running differential expression analysis. Indeed, based on the one of the largest ASD DEG datasets (<https://www.ncbi.nlm.nih.gov/pubmed/27919067>), MET is not differentially expressed between ASD and controls. The same principle holds for H3K27ac in ASD vs. controls for Figure 3c. H3K27ac signals are not corrected for potential confounders, and the P-value will not survive multiple testing correction.

11. I couldn't find any data deposited under the accession code authors provided.

12. To ensure reproducibility, I recommend that analysis codes be made fully available and referenced in the text. All data used and generated, including data generated during analysis steps, should be released or should be easy to generate from provided data and freely available codes.

Minor comments

Figure 2a, Extended Figure 4b: Astrcytes -> Astrocytes; MicroGlia -> Microglia

Reviewer #3:

Remarks to the Author:

In their study, authors explore the role of gene regulatory elements (GREs) in primate evolution based on the analysis of four species. Although the amount of data generated for the paper is minimal, authors do an excellent job in utilizing published data, delivering several novel and interesting observations.

The data analysis pipeline looks reasonable, and the effects appear to be strong enough to be trusted.

There only a few minor points I wanted to clarify:

1. Comparison between GRE gain and gene expression shown in extended data Fig. 4. It seems a bit strange to show actual expression level distributions of the genes instead of the differences (maybe as log expression fold changes) between humans and macaques. There is a range of expression values in both species, so plotting the expression differences should theoretically yield a stronger signal.

2. Term "Multidimensional scaling" usually refers to a particular dimensionality reduction algorithm different from tSNE.

Reviewers' comments:

Reviewer #1 (Remarks to the Author):

This manuscript by Castelijn et al generates new marmoset ChIP-seq and RNA-seq datasets and incorporates these data with previously generated ones from the same group from humans, chimpanzees, and macaques (Vermunt et al). The authors use these data to identify hominid (shared between humans and chimpanzees) gene regulatory elements (GREs). They analyzed prefrontal cortex and cerebellum as previously done and investigated GRE evolution using ChIP-seq for histone H3 lysine 27 acetylation. The authors also analyze H3K27ac data derived from FACS sorted nuclei from different cell types as well as publicly available H3K27ac from autism patients and healthy controls.

The authors find that the hominid GREs are notable for two reasons: 1) the gained ones are enriched in oligodendrocytes and 2) the gained ones are also overrepresented among regions that lose H3K27ac enrichment in ASD brains.

The authors also note that hominid gains are associated with regulatory regions that gain H3K27ac marks in postnatal and adult tissues compared with prenatal tissues, which further tracks with the idea that these marks are important for glia and not neuronal development which happens mostly prenatally.

In general, these results are important for our overall understanding of GRE evolution and the relative contribution of different cell types to this process. However, there are some concerns that need to be addressed.

We thank the reviewer his/her assessment and for constructive comments which we address below.

Major comments:

1) Given that the authors focus on hominid alterations, the rationale for comparison to disease-relevant data is a bit confusing. While there is certainly no way in which we could determine whether chimpanzees develop ASD, and chimps do develop pathological hallmarks of other brain disorders such as Alzheimer's disease, it seems potentially a comparison that might result in false positives. My preference would have been for the authors to define human-specific GREs and compare those to the disease-relevant datasets. In the abstract the authors make references to speciation and the human brain, so it is a bit misleading for the paper to then focus on hominid alterations. I therefore suggest modifying the abstract unless the human-specific changes will indeed be explored.

The rationale to look at neural disease comes from the functional analysis (link between sialylation and ASD) as well as literature that previously linked recent (human) evolution to neural disease. When you delve into the literature suggesting that neurodegenerative diseases recently evolved it becomes less clear whether this is a human-specific trait or something that emerged earlier. The most compelling data for the link in PD and AD is that a number of genes (SNCA, APOE) have recently changed to ameliorate neurodegeneration and some of these changes also occurred

before the split between humans and chimpanzee (SNCA). Therefore, analyzing hominins seemed reasonable to us.

Nevertheless, in the current manuscript we also assess human-specific regulatory DNA and observe that the same enrichment for hominin GREs, emerging in oligodendrocytes and being disrupted in ASD, is also seen in human-specific elements. Significance of the interaction is much lower which is likely due to the lower number of identified human-specific elements when comparing against four other primates. It is therefore possible that the regulatory changes that first emerged in hominins represent a biological system that is under continued selective pressure following the split and thus continued to evolve in humans. We have modified the abstract and have also explained this more extensively in the discussion to make this point more clear (Supplementary Fig 9c-f).

2) The hominid-specific regions (1398 gained and 532 lost) are the sum of PFC and CB hominid-specific regions. This is not really mentioned in the main text where it should be. There is a discontinuity between the main text and the figures and this might be confusing for the reader. I would be more specific regarding PFC/CB and provide specific numbers when necessary. It appears the finding related to oligodendrocytes is only relevant to data derived from the cortex and not the cerebellum? If this is the case, this should be more explicitly discussed.

We added the specific numbers of hominin gains and losses in the text at lines 92-93 to further clarify that the total number of hominin changes mentioned is the combination of changes in cerebellum and prefrontal cortex. We also mention the numbers for PFC and CB separately in Supplementary Fig. 4a and added additional analysis of CB elements (e.g Fig. 1e, 1f, Supplementary Fig 5c). The observation that hominin-specific changes indeed preferentially occur in oligodendrocytes is specific to PFC and not CB, likely because cerebellum is mostly (70-90%) granule neurons¹. This is also mentioned in lines 125-127. We have now also confirmed our observation that hominin-specific PFC gains preferentially occur in oligodendrocytes using public single cell ATAC-Seq data (Fig 2a, panel A of figure below) as well as additional FANS sorted NeuN+ and NeuN- cells (Fig 2d, panel B of figure below).

Oligodendrocyte specificity of hominin-specific PFC gains. A) bar plot showing the cell-type-specific ATAC peaks in hominin-specific gains and stable GREs. **B)** box plots

showing normalized ATAC signal on hominin-specific gains and stable GREs in both NeuN⁻ and NeuN⁺ FANS sorted nuclei.

3) It is surprising that there is no postnatal-adult enrichment of All GRE regions given that these were defined from adult individuals (Fig 2d). Can the authors comment on this a bit more? Could this be related to how these regions are identified? Maybe the regional interval ~5kb is too broad?

The reason why we do not see an overall difference between postnatal and adult stages when looking at all GREs is because there are not many differences between these stages using DESeq (see panel A of figure below). It appears that most of regulatory elements that are active in the brain at these stages support nuts and bolts processes that are required across development which may not be surprising given that in terms of cell types and structure most of the brain is there. We didn't include this in the text as it disrupted the flow. Furthermore, the differences are also supported by gene expression data showing the same thing (Fig 3b). We do not think that the average size (~4kb, Supplementary Fig 2d is in log scale) of the here identified GREs is related to this as we overlap a similar range of regulatory elements per developmental stage. Moreover, although an increase in minimum regulatory element size does increase the number of elements covered per developmental stage, this is at a similar rate for all stages (see panel B of figure below).

GREs across developmental stage. A) bar plot showing the percentage of regions differentially enriched between the indicated developmental timepoints. **B)** plot showing the number of regulatory elements per developmental timepoint covered by the here identified primate GREs with different minimum sizes.

4) Overlapping regions: what is the number of nucleotides that overlap to declare a peak overlapping?

In our analysis regions are considered to be overlapping if at least 1 base was shared between the regions. Although this seems to be lenient cut-off, any cut of is a balance between false positives and false negatives. We observe an average overlap of ~2700bp, where only 0.3% of the events is a merger between regions with less than 100bp overlap (See figure below). Thus, we do not see that our strategy creates gross anomalies in the data. This is now clarified in the methods section line 386.

Basepair overlap of merged GREs. Histogram showing the distribution of overlap size between GREs that are merged during our analysis.

5) Are biological/technical covariates taken into account?

Technical covariates such as batch effect were handled by making sure that biological replicate samples were generated in various batches at distinct time points. By increasing batch variability between replicates, we minimized consistency in batch effects ensuring these are not picked up when analyzing species. This also ensures that other samples can be added and analyzed at a later time point as we do in this manuscript. To analyze batch effect, we have performed multivariate analysis and demonstrate that batch has very little influence in the current analysis. (Supplementary Fig 3c, d and figure below).

Multivariate analysis on all GREs and for hominin-specific changes separately. Multivariate analysis was performed on the rpkm normalized H3K27ac enrichment for each GRE separately, using the stated sample characteristics as covariates.

Other confounding factors such as, sequencing platform, sequencing depth, and FRIP score also hardly influence the analysis. This is also true for biological confounders such as age, gender and postmortem delay (PMD). We find that most of the variation is explained by either the species or tissue of origin which is expected. Thus, it is unlikely major batch effect related issues are of influence on the data. This has now been incorporated in the manuscript (Supplementary Fig 3c, d).

To formally exclude batch as a confounder in the analysis, we have generated new data analyzing white matter samples across the 4 species in pre-defined batches, sequenced in a single run (see table below). This data was generated to prove that glial cell content was no major factor in the current analysis but can also serve to show that the results are not batch related. (See Fig. 2f). Multivariate analysis on these samples again demonstrates that batch has no major influence and most variation in H3K27ac enrichment is explained by the evolutionary divergence between the species (See figure below).

Table. Chip-sequencing of White matter tissue in pre-defined batches

Species	Human			Chimpanzee			Macaque			Marmoset		
Replicate	1	2	3	1	2	3	1	2	3	1	2	3
Chip-batch	A	B	C	A	B	C	A	B	C	A	B	C
Library-batch	A	B	A	B	A	B	A	B	A	B	A	B

Multivariate analysis on all GREs identified in primate white matter tissue. Multivariate analysis was performed on the rpkm normalized H3K27ac enrichment for each GRE separately, using the stated sample characteristics as covariates.

6) What is the rationale for removing the duplicates from the RNA-seq data?

The RNA-Seq libraries are total RNA libraries that are sheared as part of the protocol prior to adapter ligation and PCR. Exact replicates are therefore likely PCR duplicates and removed. We have now explained this better in the methods section.

7) The use of the NeuN- and white matter data to support the idea that the results are not simply due to alterations in glia/neuron ratios is not well explained. The authors should detail the logic behind these comparisons more clearly.

We have added extra explanation when NeuN samples are first introduced (line 127) reading: *As oligodendrocytes represent the main constituent of glial cells in the brain we used a third independent dataset² separating neurons and glial cells in PFC based on the expression of NeuN which selectively labels neural nuclei on the nuclear membrane and not glial nuclei.* We also used additional data to confirm glial identity including single cell ATAC-Seq data and new ChIP-Seq samples for white matter to confirm that hominin-specific gains are not due to an increase in glial cell content during hominin evolution.

8) The concept of taking overlaps of pairwise comparisons for the definition of hominid-specific changes does not take into account evolutionary distances of the species/parsimony. Please address this.

We suspect the reviewer is referring to the difference between human and chimpanzee in this comment. As hominids are covering all great apes, hominin is a more precise/parsimonious description of the human/chimp branch of hominids. We have changed hominid to hominin throughout the text.

Minor comments:

1) The heatmap in figure 1d might be improved if the authors scaled the gain and loss heatmaps the same way so that the difference in the numbers of gained and lost regions across tissues was more intuitive. Currently, it looks like there are equal numbers if you are only looking at the heatmaps.

We have adjusted the figure panels and scaled the heatmaps to better represent the number of regions in each set.

2) Extended figure 2f suggests that cerebellar GREs might be more divergent between hominids and monkeys than PFC GREs. Was there anything interesting from the cerebellum specific GREs to discuss?

Cerebellum is a relatively homogenous brain region, consisting mainly of granule neurons, while the prefrontal cortex consists of different neuronal cell types as well as glial cells. This usually results in a higher signal-to-noise ratio for CB compared to PFC, as can be seen from the FRiP scores (average of 40% for cerebellum and 15% for prefrontal cortex in human). Therefore, it is easier to identify significant differences in

CB than it is in PFC. We and others previously observed this for CB and PFC in ChIP-Seq data³ as well as in gene expression data⁴. We added this to the text (lines 105-107). This is also observed for the other primate species (see Supplementary table S3). To elaborate more on the CB results we have added extra analysis for cerebellum (e.g Supplementary Fig. 5c) but did not find anything standing out as clearly as when analyzing PFC.

3) How do the authors interpret the under-enrichment of gained GREs among regions with more H3K27ac signal in ASD patients in figure 3a?

Looking at the regions gained in ASD we see that these are preferentially neuronal (Supplementary Fig. 9b and figure below). This is also supported by other data^{5,6}. As a consequence, oligodendrocyte GREs may be depleted from this set.

It is likely that the link between recently evolved GREs in oligodendrocytes and ASD is either due to an oligodendrocyte transcriptional program that is affected in patients or a subtype of oligodendrocytes that is lost. The latter would result in an underrepresentation. Nevertheless, there is no clear reduction in glial cells content in ASD brains⁶⁻⁸. This is now discussed in the manuscript (lines 221-223) and (lines 270-275) in the discussion.

We also added Supplementary Fig. 9a-c to further support this.

Oligodendrocyte specific regulatory elements are affected in ASD. Metaplot analysis showing the average H3K27ac enrichment profile in different cell types for differentially enriched GREs in autism patient brains compared to healthy controls and regions identified as hominin-specific PFC gain deregulated in ASD.

4) The authors write that they removed regions that had zero rlog normalized read counts in one species. This is done prior to differential enrichment. How many GREs are removed at this step? It should also be clarified that hominid specific DE regions are compared to all GREs after this filtering step.

In total we exclude 711 out of 37,308 GREs in any of the pairwise comparisons, this is now mentioned in the text. We have adjusted the methods section to clarify that DE regions are compared to all initially identified GREs at lines 436.

5) The manuscript should be proofread as it contains many typos. Also selection of words can be improved in some sentences (e.g. “existing DNA”).

We replaced the phrase “existing DNA” with “conserved DNA” and tried our best at avoiding spelling mishaps.

6) Labels of the x-axis are wrong in Extended Data Fig3b. It should be chimpanzee not human.

We thank the reviewer for catching this and adjusted the figure legends accordingly.

7) Figure 2a: there is a typo in the y-axis. “Gians” should be “Gains”; “Astrocytes” in Ext. Data in Fig4b.

We adjusted the figure legends.

8) What is the sample size on Fig2d? In other words, how many reads from each developmental group map to hominid gains and all GRE?

We have added the requested data. The sample size is 685 hominin-specific gains for the prefrontal cortex (Fig 3a, upper left panel) and 37,308 for all GREs (Fig 3a, lower right panel). The amount of reads that map to both sets of GREs is depicted as rlog-normalized counts. For the prenatal and postnatal stages, we mapped on average 4 million reads to all GREs, of which 35,000 were mapped to hominin-specific gains. For the adult stages we mapped on average 2.6 million reads to all GREs, of which 230,000 were mapped to the hominin-specific gains.

Direct comparison of hominin-specific gains with all GREs within each developmental group shows the same increase of H3K27ac enrichment in adult tissue specifically (Supplementary Fig 7a and figure below). At prenatal and postnatal there is almost no difference in enrichment at these regions in the PFC. The hominin-specific gains are thus preferentially activated during the development between postnatal and adult stages.

Hominin-specific gains emerge postnatally. Box plots showing rlog normalized H3K27ac enrichment per developmental stage for all here identified GREs and the hominin gains in prefrontal cortex. Dissimilarities between the distributions were calculated using a Student's t-test.

Reviewer #2 (Remarks to the Author):

Castelijns et al. have compared gene regulatory elements (GREs) in monkeys versus great apes to identify hominid-specific GREs (or hominid-gains). The basic assumption is that hominid-gains will provide important information about human evolution and higher cognitive abilities of human. Hominid-gains display oligodendrocyte-specificity and are altered in ASD postmortem brains. In particular, the latter finding is in line with a series of recent papers showing that human specific evolutionary elements are linked with neuropsychiatric and neurodevelopmental disorders. However, two major claims of this paper (that they identified hominid gains and they are associated with ASD) are somewhat compromised (the datasets have potential batch effects and the link to ASD is weak), which will require significant refinement. Below I framed my concerns in more details:

We thank the reviewer for taking the time to look at our work and his/her constructive comments, which we have addressed below.

Major concerns

1. The major concern about this paper is that they used their new dataset as well as publicly available dataset (Vermunt et al), while not corrected for batches. While authors have

pointed out that the same protocol was used as Vermunt et al., batches are not only caused by the protocol, but the person who ran the experiment, reagents used for the study (e.g. whether they came from the same lot number or when they were open), the day that the sequencing was run, and many other confounders. Since both datasets were generated by the same group, is this dataset newly generated or generated at the same time as Vermunt et al.?

This type of study design may be fine if they compare peaks vs. no peak, but can be a serious problem when they run differential peak analyses (which they ran). This is a complicated issue, as batch totally corroborates with samples, and batch correction can dilute the biological signals. It is important for authors to exclude the possibility of the main finding being caused by batch effects.

We agree that batch effect can influence data analysis and are aware that they could obscure true biological signals or in cases introduce false positive results. We also agree that measures to computationally adjust for batch effects often do more harm than good to the analysis.

The way we dealt with batch effect in our previous work was to make sure that biological replicate samples were generated in various batches at distinct time points. This increase in batch variability minimizes consistency in batch effects between replicates ensuring these are not picked up, while true biological differences should remain. This strategy also ensures samples can be added and analyzed at a later time point. To illustrate this, we have performed multivariate analysis and demonstrate that batch has very little influence in the current analysis. This is true for all elements as well as hominin-specific changes. This has now been incorporated in the manuscript (Supplementary Fig. 3c, d and figure below).

Multivariate analysis on all GREs and for hominin-specific changes separately. Multivariate analysis was performed on the rpkms normalized H3K27ac enrichment for each GRE separately, using the stated sample characteristics as covariates

Other confounding factors such as sequencing platform, sequencing depth, and FRIP score also hardly influence the analysis. This is also true for biological confounders such as age, gender and postmortem delay (PMD). We find that most of the variation is explained by either species or tissue of origin which is expected. Thus, it is unlikely major batch effect related issues are of influence on the data. This has now been incorporated in the manuscript (Supplementary Figure 3c, d).

To formally exclude batch as a confounder in the analysis, we have generated new data analyzing white matter samples across the 4 species in pre-defined batches, sequenced in a single run (see table below). This data was generated to prove that glial cell content was no major factor in the current analysis but can also serve to show that the results are not batch related (see Fig. 2f). Multivariate analysis on these samples again demonstrates that batch has no major influence and most variation in H3K27ac enrichment is explained by the evolutionary divergence between the species (See figure below).

Table. Chip-sequencing of White matter tissue in pre-defined batches

Species	Human			Chimpanzee			Macaque			Marmoset		
Replicate	1	2	3	1	2	3	1	2	3	1	2	3
Chip-batch	A	B	C	A	B	C	A	B	C	A	B	C
Library-batch	A	B	A	B	A	B	A	B	A	B	A	B

Multivariate analysis on all GREs identified in primate white matter tissue. Multivariate analysis was performed on the rpkms normalized H3K27ac enrichment for each GRE separately, using the stated sample characteristics as covariates

2. Authors have not mentioned about whether they regressed out any confounding variables while running DESeq2. Comprehensive codes they used to perform the analysis will be helpful (please also see comment #12).

The current analysis has been done with 3 biological replicates per primate group in a pairwise fashion, which is not enough to properly regress out confounding variables using DESeq2 for which preferentially many more samples per experimental group are needed. This is not feasible, especially with rare primate tissue. Based on the multivariate analysis, as described above, we concluded that most variation within our data set was of biological origin (e.g. species or tissue) and that technical variation (e.g. library depth and FRIP score) had only a minor influence. We therefore ran DESeq2 on the raw H3K27ac read counts and relied on its internal normalization procedure which adjusts counts for library size, sample quality and bias towards highly enriched elements.

We have not generated custom code, as further elaborated in response to comment #12, and have run DESeq2 according to the user manual (https://bioconductor.org/packages/release/bioc/html/DESeq2.html).

3. Rather than putting PC1 and PC2 separately for Figure 1c, can authors put them in the same plot as Figure 1b? This will give a clearer clustering pattern among all groups.

We adjusted the plot in Figure 1c and combined PC1 and PC2 in one plot.

4. How many peaks did they find that are not changing across evolution (detected for all four species = stable GRE)? A more comprehensive picture will arise if authors can provide # of peaks that fall onto every sector of the Venn diagram in Extended Data Fig. 3c.

A stable GRE is defined as not DE in any of the pairwise comparisons. We annotate 22,304 and 14,203 stable regions for prefrontal cortex and cerebellum respectively. To be extra stringent we took the overlap between those two and used the resulting 8,441 regions as stable across primate evolution in both brain regions. The numbers of either species-specific gains or losses are added to the appropriate sectors of the venn diagram (see figure below). We adjusted Supplementary Fig 4a accordingly.

Evolutionary changes at regulatory elements. Venn diagrams showing hominin-specific regulatory changes (highlighted in yellow) and species-specific gains and losses as defined by differential enrichment analysis with DESeq2. Numbers indicate the number of GREs in each category.

5. Can authors repeat Figure 2a-b and Figure 2d for stable GREs? It would be informative to see how species-specific elements differ compared to the ones that do not change across evolution. Also, do authors have explanation of why microglia are depleted of any GREs in Figure 2b? Is this because of the quality of H3K27ac datasets in microglia?

We thank the reviewer for catching the depletion of microglia signal. The reason for this was indeed related to sample quality (FRIP 2.7%). We replaced the sample with another replicate from the same study with a better signal-to-background ratio (FRIP 44.2%). We redid the analysis and adjusted the figures accordingly (see figure below, panels a-b). We have also added the requested figures for stable elements (Fig 2c, Fig. 3a and figure below), where no enrichment for oligodendrocyte specific peaks was observed. However, we do see a tendency for neuronal peaks to be overrepresented in stable GREs which could indicate that neuronal function is relatively better conserved across primate evolution.

Cell type specificity of primate GREs. A) heatmap showing H3K27ac enrichment for 685 scaled hominin-specific PFC gains, analyzed in different cell types as indicated. Heatmap colors indicate H3K27ac enrichment (rpm). **B)** metaplot analysis showing the average H3K27ac profile in different cell types for hominin-specific PFC gains, stable GREs and all GREs. **C)** box plots showing normalized H3K27ac enrichment in prefrontal cortex samples across developmental timepoints for hominin-specific PFC gains, stable GREs and all GREs. Dissimilarities between distributions were calculated using a Student's t-test.

Analysis of stable GREs across development did not recapitulate the observed enrichment of hominin-specific gains in adult elements (Figure above, panel C). We observed some variation between developmental stages for stable elements although the effect size was marginal especially when compared to the enrichment of hominin-specific PFC gains in adult stages.

6. It is hard to compare Figure 2b results between hominid-gains and all GREs because the scales differ, but according to Extended Figure 4c, it seems that NeuN+ read counts are smaller for hominid-gains than all GREs. If this is really true, does it mean that hominid-gains are under-represented for neuronal peaks?

In order to make the comparison between the enrichment profiles easier, we adjusted the scales of Fig 2c. Hominin-specific gains in the prefrontal cortex are indeed depleted for H3K27ac in NeuN+ sorted nuclei as indicated in Supplementary Fig. 5d, e. Most gains are oligodendrocyte derived and since the signal is an average across all identified gains, as a result NeuN+ signal drops.

To assess relative cell numbers and not signal we used single cell ATAC-Seq data to prioritize regions towards a single cell type and find that neuronal peaks are still present in our hominin-specific gains however only a small proportion of these regions originate specifically from neuronal cell types (Fig. 2a, d and figure below).

Oligodendrocyte specificity of hominin-specific PFC gains. A) bar plot showing the cell-type-specific ATAC peaks in hominin-specific gains and stable GREs. **B)** box plots showing normalized ATAC signal on hominin-specific gains and stable GREs in both NeuN⁻ and NeuN⁺ FANS sorted nuclei.

7. Currently the basic genomic composition of hominid gains/stable GREs is unclear. How many of them overlap with promoters and exons? How many of them are located in the non-coding region (intronic/intergenic)?

We have added the requested information in Supplementary Table 5 (see table below). We find that stable GREs are more often located at TSSs (2 fold enriched) and hominin-specific gains and losses in both PFC and CB are more often distal, which is expected as promoter distal regulatory elements show more rapid evolutionary turnover⁹. Both hominin-specific gains at enhancers and hominin-specific gains at promoters show an enrichment of H3K27ac signal in oligodendrocytes (See figure below). Losses are more often intergenic, both for PFC and CB, which is an interesting observation on itself. It is possible that intragenic enhancers are more conserved as they co-evolve with the genes in which they are located.

Table. Genomic composition of selected GREs

	# regions	# TSS	# exonic	# intronic	# intergenic
--	-----------	-------	----------	------------	--------------

All GREs	37308	9292	6857	11411	9748
PFC hominin gains	685	53	163	264	205
PFC hominin losses	158	15	17	52	74
CB hominin gains	713	101	216	187	209
CB hominin losses	274	26	68	147	133
Stable GREs	8441	3781	1027	2167	1466

Cell type composition of hominin-specific PFC gains. Metaplot analysis showing the average H3K27ac profile in different cell types for hominin-specific PFC gains separated based on overlap with known human TSSs.

If the majority of them are intergenic and intronic, given that H3K27ac marks enhancers that often regulate distal genes via long-range interactions, I recommend authors to identify putative target genes based chromatin interactions, and repeat the analysis in Figure 2e and Extended Data Figure 4g. Authors have already done this in Figure 3f for a MET locus. As chromatin contact profiles in the brain tissue (<https://www.ncbi.nlm.nih.gov/pubmed/30545857>) and brain cell types (<https://www.ncbi.nlm.nih.gov/pubmed/30545851>) now became available, they can perform a similar analysis in a genome-wide fashion. There's also a paper that linked regulatory elements reported by Vermunt et al. to their target genes using Hi-C data: <https://www.ncbi.nlm.nih.gov/pubmed/31160561>.

We agree that the analysis of optimal HiC datasets is preferable to linking genes by proximity, which misses interactions with multiple genes and instances were enhancers skip genes. The latter was in part omitted by us by using closest active genes as a proxy. However, many of the HiC datasets are still not sequenced to a high enough depth to provide a strong enough resolution and thus have a high false negative rate.

Furthermore, as loops are also cell-type-specific, we would be looking for an adult oligodendrocyte dataset while the datasets proposed are early developmental

neural cell types exposed to tissue culture. These data are now mentioned in our manuscript.

We did analyze the first dataset proposed by the reviewer as it represents chromosomal interaction in the prefrontal cortex. The downside of this dataset is that cortex is heterogeneous which makes finding cell-type-specific interactions even harder. For instance, c-MET was not found to be interacting with the closely located hominin-specific PFC gains in the HiC data but did show interaction with both the 3' and the 5' GRE using 4C from the promoter (See figure below Panel A). Nevertheless, the interactions that were found were added to our analysis. 6,350 GREs were linked to a TSS using the PFC HiC data, while 14,262 out of 28,016 enhancers were not represented. To incorporate these data, we used the HiC defined enhancer-promoter interactions to override our proximity calls. If a HiC interaction was not found, we coupled based on proximity. Using this updated gene-assignment we repeated the analysis in Fig. 3b and the analysis of Supplementary Fig. 7d (see figure below). Analysis of gene expression difference between human and macaque for genes linked to either hominin gains or stable GRE revealed a similar trend for genes based on proximity or on HiC gene-assignments (panel B). Similarly, we observed the same elevated human expression in adult tissue compared to pre- and post-natal stages (panel C).

Enhancer-promoter linkage for hominin-specific PFC gains. A) 4C-seq and H3K27ac ChIP-seq track of human white matter of a region containing the c-MET gene. The c-MET promoter was used as viewpoint in the 4C analysis. The green highlighted areas indicate hominin-specific PFC gains. Chromosomal interactions, as defined by HiC in prefrontal cortex, are depicted below. **B)** box plots showing gene expression

differences between human and rhesus macaque for either hominin-specific PFC gains and stable GREs. GREs were assigned to target genes based on either proximity or with incorporation of the HiC data. Dissimilarities between distributions were calculated using a Student's t-test. **C)** box plots showing gene expression differences between human and rhesus macaque at different developmental timepoints for genes linked to hominin-specific PFC gains by incorporation of HiC data. Dissimilarities between distributions were calculated using a Student's t-test.

We replaced the panels of Fig. 3b with the above displayed panels and adjusted to text and methods to incorporate the above stated linkage procedure and added references of the suggested papers.

8. It seems that GOs enriched for hominid-specific changes in the PFC and Cerebellum mark widely different processes. I couldn't find Figure 1e-f and 2a-e equivalent for hominid-gains in the cerebellum. Do hominid-gains in cerebellum exhibit different cell-type specificity? What's authors explanation to this?

Cerebellum and prefrontal cortex have completely different cellular compositions. Where the prefrontal cortex consists of many different neuronal and non-neuronal cell types, the cerebellum consists of ~80% densely packed granular neurons. Thus, finding oligodendrocyte peaks in cerebellum is unlikely. This explains why the hominin-specific changes are associated with different cellular processes (and cell types) in both tissues. We have added extra analysis for the hominin-specific changes in cerebellum in the manuscript and discuss the difference in composition (e.g Fig. 1e,f, Supplementary Fig. 5c, see figures below).

Analysis of hominin-specific cerebellum gains. A) conservation (20 mammals) using phastCon scores as defined by UCSC of hominin-specific CB gains compared to stable GREs. Dissimilarity between the distribution was calculated using a Student's t-test. **B)** analysis as in **A** for hominin-specific nucleotide changes. **C)** metaplot of average H3K27ac profile in different cell types for hominin-specific CB gains.

As was shown for prefrontal cortex gains, hominin-specific gains in the cerebellum were genetically less conserved, as indicated by the lower phastCon score and increased hominin-specific DNA changes (Fig 1e, f and panel A-B in figure above). The difference in cell type composition is also reflected in the cell-type-specific H3K27ac signals for the hominin-specific CB gains, these regions do not show specific enrichment for cell types mainly found in the prefrontal cortex (see panel C in figure above). Moreover, we observed a reduction of H3K27ac signal in CB for PFC-gains and vice versa, indicative for the difference in cell-type content between the two regions and the cell type specificity of these regions (See figure below).

H3K27ac enrichment on hominin-specific gains. Box plots showing the normalized H3K27ac enrichment in cerebellum and prefrontal cortex for hominin-specific gains and stable GREs.

9. Extended Data Figure 5b is interesting as it seems that a hominid-gain is linked to the same gene in all four species. How often this is the case that hominid-gains are linked to the same gene in all species? Is there a case that they mark different genes in different species?

The linkage between a GRE and its target gene was done based on the UCSC hg38 RefSeq list by proximity. The reason for this is that many genes are not well annotated across the four species and thus we used the best genome to link enhancers to genes (see Supplementary Fig 7b, c). To assess to what extent this could lead to misclassification of target genes to GREs in the primate species, we compared the macaque RefSeq gene (rheMac8) list with the human TSSs mapped to macaque genome.

For instance, the UCSC RefSeq list of rhesus macaque consists of only 6460 genes, of which 5804 (90%) overlap with the same human gene when lifted to the macaque genome (see panel A of figure below). In addition, liftover of human TSS regions to the macaque genome, shows that most are enriched for H3K4me3 in the macaque brain, indicating that they mark potential promoter regions on the macaque

genome (see panel A of figure below). Assessment of the remaining macaque TSSs shows that many are located close to the human projecting TSS and could be alternative start sites in humans (see panel B-C of figure below). Thus, we believe that species-specific misclassification of gene promoters because a gene moved is minor, especially considering other issues with proximity-based linkage as pointed out by the reviewer.

Promoter conservation across the primate lineage. A) ChIP-seq tracks showing rpm normalized H3K4me3 enrichment of cerebellum and prefrontal cortex of rhesus macaque. Rhesus macaque TSS as defined by UCSC RefSeq and projected Human TSS are depicted below. **B-C)** Chip-seq tracks showing the rpm normalized H3K4me3 and H3K27ac enrichment of cerebellum and prefrontal cortex in rhesus macaque (**B**) and human (**C**). TSS as defined by UCSC RefSeq are depicted below, as well as the primate GREs identified in this study.

10. Another major concern is that the link between hominid-gains and ASD is weak.

10-1. First, there was no enrichment of ASD GWAS signals in hominid-gains. This can be partly because authors used lead SNPs from the NIH GRASP, which has a limited number of hits for ASD GWAS. Therefore, I strongly encourage authors to use stratified LD score regression (<https://www.ncbi.nlm.nih.gov/pubmed/26414678>) to correct for LD and take genome-wide signals into account.

We have performed stratified LD score regression to assess whether disease heritability is enriched in the hominin-specific regions, using the LDSC software as described in the mentioned paper by Finucane et al (<https://github.com/bulik/ldsc>). Summary statistics of GWAS for Alzheimer's Disease, Autism Spectrum Disorder, Parkinson's Disease, Bipolar Disorder and Schizophrenia were obtained and both the GREs identified here as well as the autism-specific regulatory changes were added to the baseline model of Finucane HK et al. Stratified LD score regression on these elements showed that, while GREs and in particular promoter regions are enriched for schizophrenia variants (see panel A figure below), we did not observe significant

enrichment in heritability at hominin-specific regions for any of the analyses (see Supplementary Fig. 8a, b panel B figure below). However, finding an interaction in this manner is also not straightforward due to the low number of recently evolved loci, where LDSC analysis is known to have problems calculating heritability.

To assess whether extension of the regions would allow for a better heritability calculation we extended the hominin-specific changes to 25kb and added these regions to the baseline (see insert of panel B figure below). Although extension yielded more robust results and less negative heritability scores, none of the hominin-specific changes were enriched for disease heritability. Indicating that these regions are not associated with a disease phenotype solely based on genetic variation.

As we agree with the reviewer that stratified LD score regression is a more robust method to assess SNP enrichment compared overlap with GRAPS lead SNPs alone, we have replaced the lower panel of Fig. 4a with panel C of the figure below (see Supplementary Fig. 8a). Next to the enrichment of the here identified regions, we analyzed the autism-specific regulatory elements as defined by Sum W et al. and were able to replicate their finding that these regions are linked to psychiatric disorder SNPs.

Stratified LD score regression. A) bar plot showing the enrichment of disease heritability for the in our manuscript identified GREs, divided into several subsets. Colors indicate the heritability enrichment per disease. (AD: Alzheimer's Disease, ASD: Autism Spectrum Disorder, BIP: Bipolar Disorder, PD: Parkinson's Disorder, SCZ: Schizophrenia). **B)** bar plot as in **A** for the hominin-specific regulatory changes. The insert shows the heritability enrichment score of hominin-specific prefrontal cortex

gains extended to 25kb. C) heatmap showing the enrichment score of different disease heritability for different sets of regulatory elements. Only the significant enrichments are displays, number indicate p-value, color scale indicates the enrichment score (CB: cerebellum, PFC: prefrontal cortex, GRE: gene regulatory element).

10-2. Second, ASD H3K27ac data that authors used <https://www.ncbi.nlm.nih.gov/pubmed/27863250> often display opposite direction to what we would expect from the expression data (e.g. neuronal genes were downregulated in ASD postmortem brains, while H3K27ac peaks were upregulated). Consequently, it is currently unclear whether ASD differential H3K27ac peaks mark causal or responsive processes.

We do not believe the data that the reviewer referred to is an indication of indirect effects as there are many reasons why neuronal peaks could be up and neuronal genes down in ASD brain. Neuronal is in this case indirectly defined based on GO analysis and these terms typically represent a handful of genes that are overrepresented. Thus, it is not clear from the work that the reviewer refers to whether these were the same genes.

Furthermore, the effect of altered GREs on genes may not always be straightforward as enhancer changes can be expression neutral through compensation (either by another enhancer changing in the opposite direction or by shadow enhancers that function as a backup). Worse than this, GREs that increase activity can compete with other stronger enhancers resulting in downregulation of expression, while intragenic GREs can physically interfere with transcription. Thus, while enhancer gains generally result in a modest enhancement of gene expression, we should really be assessing them in the context of altered gene regulation as opposed to enhanced gene regulation.

Nevertheless, we agree with the reviewer that many altered genes as well as H3K27ac peaks may be responsive rather than causal which can still be informative on what the disease eventually represents in terms of altered physiology. It is thus very well possible that the changes in ASD observed are the result of either a gene expression program gone awry or a cellular subtype that is over or underrepresented (discussed below) as it doesn't make sense for large groups of enhancers to acquire parallel mutations in a single patient.

Moreover, ASD-upregulated H3K27ac peaks were enriched for neuronal signals, while ASD-downregulated H3K27ac peaks were largely enriched for glial signals. Therefore, there is a chance that the enrichment in Figure 3a just captures cell-type specificity of ASD H3K27ac signals (e.g. ASD-downregulated H3K27 peaks largely mark oligodendrocyte signals, which was consequently captured via hominid-gains).

Therefore, additional layer of evidence will be necessary to support this claim. A few suggestions I have are the followings: Once authors identify the target genes of hominid-gains (as suggested in comment #7), they can check whether those genes are enriched for ASD risk genes.

While it is likely that a proportion of the interaction between ASD and hominin-specific PFC gains is driven by both affecting oligodendrocytes, LLM3D analysis¹⁰ demonstrates that this is not the sole reason for the interaction (Line 233). Enrichment for oligodendrocyte specificity of the shared regions is substantially more than expected for a random interaction between the two sets.

We also do not believe that oligodendrocyte content is the main reason for this interaction as not all oligodendrocyte-specific regions are affected. We added Fig 4f and Supplementary Fig. 9a, b as further support for this. We cannot exclude that a specific subset or subtype of oligodendrocytes (that is characterized by hominin changes) is not lost in ASD brains. Furthermore, it is possible that a (recently evolved) oligodendrocyte transcriptional program is preferentially affected in ASD. Both would be interesting explanations for our observations and are now discussed in the manuscript.

As suggested by the reviewer we also looked at ASD risk genes. However, as the number of genes linked to hominin changes and ASD is not that high (n = 78), we do find known risk genes for autism linked by both HiC and proximity to the here identified hominin-specific gains that are deregulated in ASD, including c-MET, AGAP1, UBE2H and CAMK2A (see figure below). The latter locus is now discussed in the manuscript (Supplementary Fig. 8c).

Hominin-specific gains link to autism risk genes. ChIP-seq tracks showing rpm normalized H3K27ac enrichment in the PFC of primate species as indicated at loci containing the AGAP1 genes (upper left), UBE2H gene (upper right) and CAMK2A gene (bottom). Hominin-specific PFC gains are highlighted in green. Chromosomal interactions, as defined by HiC in PFC, are depicted below.

Authors can also repeat the process with other brain disorder-relevant H3K27ac signals (e.g. Alzheimer’s disease; <https://www.ncbi.nlm.nih.gov/pubmed/30349106>), which will provide disease-specificity of the enrichment. I believe there will be also other ways to bolden the claim.

We have analyzed the H3K27ac changes specific to Alzheimer's disease, as identified in the paper from Marzi et al, 2018. We do not observe the same changes that are observed with ASD. Instead we observe that the hominin-specific gains are depleted of regions that are lost in Alzheimer patients (Fig. 4a and panel A of figure below). As our hominin-specific PFC gains originate mainly from oligodendrocytes (see Fig. 2a-c) it is no surprise that these regions are depleted from AD losses as these are expected to be neuronal. We further verified that regions lost in AD are mainly of neuronal origin by assessing the cell-type-specific H3K27ac signal on these regions (Supplementary Fig. 9b and panel B of figure below). Assessing H3K27ac signal of neuronal and glial cell types reveals that regions lost in AD are enriched in glutamatergic and GABAergic neurons, while regions that gain H3K27ac in AD are enriched for oligodendrocytes and astrocytes.

Alzheimer's disease specific regulatory changes. A) heatmap showing the enrichment of disease associated GREs measured to be deregulated in autism spectrum disorder and Alzheimer's disease patient brains compared to healthy controls in correlation to hominin-specific changes found in CB and PFC. Color coding indicates odd ratio of enrichment compared to all identified GREs. P-values were determined using a Fisher's exact test and represented in $p < 0.01$. **B)** metaplot analysis of the average H3K27ac enrichment profile in different cell types for GREs differentially enriched in the entorhinal cortex of Alzheimer's disease patients compared to healthy controls and all identified GREs in both Alzheimer's disease patients and control samples.

10-3. Third, while the level of convergence of functional genomic evidence at the MET locus is remarkable (4C experiment, GRE H3K27ac vs. MET gene expression, GRE H3K27ac vs. MET promoter H3K27ac), its link to ASD is not very promising. For example, I don't think MET expression differences between control and patients will survive multiple testing correction of a typical DEG analysis. Authors neither corrected for potential confounders such as RIN,

batches, and PMI when running differential expression analysis. Indeed, based on the one of the largest ASD DEG datasets (<https://www.ncbi.nlm.nih.gov/pubmed/27919067>), MET is not differentially expressed between ASD and controls. The same principle holds for H3K27ac in ASD vs. controls for Figure 3c. H3K27ac signals are not corrected for potential confounders, and the P-value will not survive multiple testing correction.

We appreciate the reviewer's sentiment and further in-depth research is required to figure out what exactly is going on at the c-MET locus. However, we do not agree with the notion that p-values that do not survive multiple testing mean that no interesting biology is present at a locus. There are many examples where multiple testing and correction for confounders obscures signal that is biologically relevant in large scale analysis. As such p-values were never intended to be used as a truth stamp. In this regard, we did not arrive at the c-MET locus based on the data that the reviewer is referring to alone and our data is not the only data pointing towards a role for c-MET in ASD.

The effects of the enhancers could be context specific, very subtle or relevant to a subset of patients. As there are plenty of examples where subtle gene expression changes result in relevant phenotypes and patient heterogeneity is a given in ASD, I would not discount c-MET as a promising ASD gene yet based on the patient population tested as a whole.

It is possible, judging from HiC data, that the enhancers span a sub-TAD region. How these cooperate and regulate expression of c-MET and whether there is patient specificity needs to be further investigated. We now offer a more balanced discussion of the data on c-MET and also discuss an additional region where hominin-specific gains are linked to autism risk genes (see answer to comment #10.2 and Supplemental Fig. 8c).

11. I couldn't find any data deposited under the accession code authors provided.

We apologize for not providing a link and reviewer token to the here generated data. All data generated this study is deposited at GEO with accession: GSE130871 (<https://www.ncbi.nlm.nih.gov/geo/query/acc.cgi?acc=GSE130871>). The reviewer token for access is: c1mvoowiflmpfkz

12. To ensure reproducibility, I recommend that analysis codes be made fully available and referenced in the text. All data used and generated, including data generated during analysis steps, should be released or should be easy to generate from provided data and freely available codes.

We have not generated custom code for this analysis even though our disclaimer would suggest so. All code used for the analysis is freely available and referenced to in the methods section.

Minor comments

Figure 2a, Extended Figure 4b: Astrocytes -> Astrocytes; MicroGlia -> Microglia

We thank the reviewer for catching this and adjusted the legends accordingly.

Reviewer #3 (Remarks to the Author):

In their study, authors explore the role of gene regulatory elements (GREs) in primate evolution based on the analysis of four species. Although the amount of data generated for the paper is minimal, authors do an excellent job in utilizing published data, delivering several novel and interesting observations.

The data analysis pipeline looks reasonable, and the effects appear to be strong enough to be trusted.

We thank the reviewer for his/her favorable assessment of our work and have addressed the comments below.

There only a few minor points I wanted to clarify:

1. Comparison between GRE gain and gene expression shown in extended data Fig. 4. It seems a bit strange to show actual expression level distributions of the genes instead of the differences (maybe as log expression fold changes) between humans and macaques. There is a range of expression values in both species, so plotting the expression differences should theoretically yield a stronger signal.

We agree with the reviewer that log fold changes are a more regular way of displaying gene expression differences and have therefore replaced Supplementary Fig. 7d with the figure below.

Gene expression across the primate lineage. Box plots showing the difference in expression between human and macaque for genes associated with hominin-specific PFC and CB gains and stable GREs. Dissimilarity between the distribution was calculated using a Student's t-test.

2. Term “Multidimensional scaling” usually refers to a particular dimensionality reduction algorithm different from tSNE.

We adjusted the text at line 81 to “Dimension reduction and visualization with t-SNE” to clarify that the t-SNE algorithm was used for visualization of the data in 2-dimensional space.

References:

1. Andersen, B. B., Korbo, L. & Pakkenberg, B. A quantitative study of the human cerebellum with unbiased stereological techniques. *J. Comp. Neurol.* **326**, 549–560 (1992).
2. Girdhar, K. *et al.* Cell-specific histone modification maps in the human frontal lobe link schizophrenia risk to the neuronal epigenome. *Nat. Neurosci.* **21**, (2018).
3. Vermunt, M. W. *et al.* Epigenomic annotation of gene regulatory alterations during evolution of the primate brain. *Nat. Neurosci.* **19**, (2016).
4. Brawand, D. *et al.* The evolution of gene expression levels in mammalian organs. *Nature* **478**, 343–348 (2011).
5. Grove, J. *et al.* Identification of common genetic risk variants for autism spectrum disorder. *Nat. Genet.* **51**, 431–444 (2019).
6. Velmeshev, D. *et al.* Single-cell genomics identifies cell type-specific molecular changes in autism. *Science (80-.)*. **364**, 685–689 (2019).
7. Zikopoulos, B. & Barbas, H. Changes in prefrontal axons may disrupt the network in autism. *J. Neurosci.* **30**, 14595–609 (2010).
8. Donovan, A. P. A. & Basson, M. A. The neuroanatomy of autism - a developmental perspective. *J. Anat.* **230**, 4–15 (2017).
9. Villar, D. *et al.* Enhancer Evolution across 20 Mammalian Species. *Cell* **160**, 554–566 (2015).
10. Geeven, G. *et al.* LLM3D: a log-linear modeling-based method to predict functional gene regulatory interactions from genome-wide expression data. *Nucleic Acids Res.* **39**, 5313–5327 (2011).

Reviewers' Comments:

Reviewer #1:

Remarks to the Author:

In general, the authors have been mostly responsive to my concerns but I don't think all of the questions have been adequately addressed.

1) Overlapping regions: the response to this question is not entirely satisfactory. The authors now state that the overlap is 1bp but that the majority of the peaks overlap with more >1800ish bp (based on the plot). It would have been preferable if the overlap was done by percentage (>50% of the peaks should overlap) not by bp. I would use a percentage of overlap rather than number of bp overlap even though the average size of the GRE is 4kb and repeat the analysis. Or at least prove the point that with this approach the overall story will not change.

2) The covariates and batch effect concern was not clearly answered. The authors provide a variance explained plot (where one can see that batch has a median of ~10% of variance explained and it is the major covariate in gains/losses Hominin plot). The authors say that the sample size is not enough (3 samples) and therefore that is why they did not include any of the covariates in the analysis. As it stands, the concern is not answered since there is an influence even given the small N. The authors did not apply any statistics to consider the covariates (e.g. MARS) and they did not compare with/without covariates in their modeling.

3) It is impossible to say if these changes are hominid specific (most great apes) or hominin specific (human and chimpanzee) and the authors also switch this between papers. The introduction should mention what changes these exactly are (likely evolved before human-chimpanzee and after old-world monkeys are separated) and let the reader know that they are henceforth referred as hominin specific for simplicity.

New comments:

4) How exactly did the authors overlap cell-type specific peaks and GREs of this paper? How did they define oligodendrocyte specific peaks in this data? Did they perform differential accessibility per cluster and overlap the DA peaks with GREs? I could not see any description about this in the methods section. Please provide more detailed methods on how the scTHSseq data was analyzed. In some cases the authors use FANS sorted Sox10+ data or white matter, but not in all cases. Nevertheless, in no case are the methods detailed.

5) Fig 4a shows significant overlap between Hominin gains and ASD up as well as ASD down. The authors only focus on ASD down in Fig4f. It is not clear why they don't show ASD down up too.

6) What do the results (especially Fig2 and 3) look like when you include - or separately perform it with- hominin losses? This could at least be mentioned in the text since the authors claim that losses are more likely to be biologically relevant and selected against.

Reviewer #2:

Remarks to the Author:

I appreciate that authors have addressed a lot of my comments and added substantial amount of work during the revision. My major question on the batch correction has been resolved, and the link to oligodendrocytes is now robust. However, the link to ASD is still not convincing, as the majority of orthogonal validation neither yielded consistent nor significant results. Currently the only support they got related to ASD was the overlap with differential H3K27ac in ASD postmortem brains, while I am not convinced that this necessarily represents any causal link to ASD.

Given that authors used c-MET locus as an example of the ASD locus, this gene was neither captured from the largest ASD exome sequencing (Satterstrom et al., bioRxiv; Ruzzo et al., Cell) nor GWAS (Grove et al., Nat Genet). I am wondering whether they can find additional evidence with other ASD risk genes. They have named a few, but none of them seems to be the high-confidence candidate ASD genes.

Reviewer #3:

Remarks to the Author:

I'm happy with authors' response to my comments. I have no additional questions or comments.

Reviewer #1 (Remarks to the Author):

In general, the authors have been mostly responsive to my concerns but I don't think all of the questions have been adequately addressed.

1) Overlapping regions: the response to this question is not entirely satisfactory. The authors now state that the overlap is 1bp but that the majority of the peaks overlap with more >1800ish bp (based on the plot). It would have been preferable if the overlap was done by percentage (>50% of the peaks should overlap) not by bp. I would use a percentage of overlap rather than number of bp overlap even though the average size of the GRE is 4kb and repeat the analysis. Or at least prove the point that with this approach the overall story will not change.

The reviewer asked to specify the number of nucleotides per overlap, which we provided. Now the reviewer is asking for a percentage. Throughout the manuscript we consider peaks to be overlapping if a single base pair is shared. Although we agree that in theory this may result more often in separate peaks being merged, any cutoff chosen will result in a balance between wanted mergers that are missed and incorrect mergers that were performed. We feel it is important not to miss wanted mergers because it is harder to call differentially enriched regions that are large and the result of unwanted mergers. These are thus less likely to contribute to our analysis.

To show that our findings are not affected by this approach, we demonstrate that only a minor fraction of the here identified regions is a potential merger between regions that reciprocally overlap with less than 50% (1365 from 37308 GREs) (see panel a of figure below). When we exclude these regions from the further analysis, we observe no change in the oligodendrocyte specificity of hominin-specific gains, nor in their overlap with regions deregulated in autism (see panel b-c of figure below). We did not include this in the manuscript but add an explanation as to why we chose 1bp to the methods section.

Overlap of GREs. a) Histogram showing the distribution of overlap size as ratio of the GRE size between GREs that are merged during our analysis, in 10% bins. **b)** bar plot showing the distribution of cell-type-specific ATAC peaks in hominin-specific gains and stable GREs for both our initial analysis (shaded bars) or when only using regions that originate from mergers where regions share more than 50% of their

bases (full color bars). **c)** heatmap showing the enrichment for disease associated GREs measured to be deregulated in autism spectrum disorder (ASD) or Alzheimer's disease (AD) patients brains in correlation with hominin-specific changes found in cerebellum (CB) and prefrontal cortex (PFC). Only using those GREs that originate from a merger where regions share more than 50% of their bases. Color indicated odds ratio of enrichment compared to all GREs. p-values were determined using a Fisher's exact test and represented in the heatmap if $p < 0.01$.

2) The covariates and batch effect concern was not clearly answered. The authors provide a variance explained plot (where one can see that batch has a median of ~10% of variance explained and it is the major covariate in gains/losses Hominin plot). The authors say that the sample size is not enough (3 samples) and therefore that is why they did not include any of the covariates in the analysis. As it stands, the concern is not answered since there is an influence even given the small N. The authors did not apply any statistics to consider the covariates (e.g. MARS) and they did not compare with/without covariates in their modeling.

As stated in our previous revision, multivariate regression doesn't work well with low N, especially if there are more variables than samples, which may result in overfitting of the data. Furthermore, it is not possible to correct for batch in our initial data as each sample is a different batch. Correcting for this will make all samples the same, even though they are from different species/tissues.

Therefore, in our revised manuscript we did not only provide a variance plot, we also generated new samples that are batch controlled. These batch-controlled sets still enrich for oligodendrocytes and still link to regions that are deregulated in ASD. This analysis is now added (Supplementary Fig. 9g, h) and proves that batch is not the reason for our observations.

If we do try to regress out batch from this batch-controlled set, we lose a third of our regions which is a testimony to the overfitting problem. Nevertheless, the regions that are left are still enriched for oligodendrocytes and still significantly link to regions deregulated in ASD (see below) again proving that batch is not the reason for our observations. We do not include this in the manuscript as we believe this analysis is not appropriate.

Batch corrected hominin changes. a) heatmap showing enrichment of disease associated GREs measured to be deregulated in autism spectrum disorder (ASD) and

Alzheimer's disease (AD) patient brains in correlation with hominin-specific white matter changes. Color coding indicates odds ratio of enrichment compared to all identified white matter GREs. p-values were determined using a Fisher's exact test and represented in the heatmap if $p < 0.01$.

In addition, we analyzed the variance plot that was generated using multi variate linear regression and selected regions that were significantly associated with confounders. As DESeq2 already controls for sequencing depth variations, we flagged all regions where either PMD, batch, gender or sequencing platform were significantly contributing (See panel c of figure below) and removed these from the analysis (1495 out of 37308 GREs). We observed that only a few hominin-specific changes are affected (27 out of 1930 hominin changes of which only 1 is a hominin gain that is also deregulated in ASD) and that the overall conclusions do not change. This is mentioned in the methods section but not employed in the analysis as we do not feel that regression on low N is appropriate.

Covariate analysis. Violin plots showing the p-value of covariate contribution as assessed by multivariate linear regression using ANOVA on all GREs. Dashed line indicates the significance threshold $p < 0.01$.

3) It is impossible to say if these changes are hominid specific (most great apes) or hominin specific (human and chimpanzee) and the authors also switch this between papers. The introduction should mention what changes these exactly are (likely evolved before human-chimpanzee and after old-world monkeys are separated) and let the reader know that they are henceforth referred as hominin specific for simplicity.

We have clarified this in the introduction line 52

New comments:

4) How exactly did the authors overlap cell-type specific peaks and GREs of this paper?

How did they define oligodendrocyte specific peaks in this data? Did they perform differential accessibility per cluster and overlap the DA peaks with GREs? I could not see any description about this in the methods section. Please provide more detailed methods on how the scTHSseq data was analyzed. In some cases the authors use FANS sorted Sox10+ data or white matter, but not in all cases. Nevertheless, in no case are the methods detailed.

We have provided a more extensive explanation on how cell type specificity was determined in the methods section. In short, we used the list of differentially enriched region per cell type, made available by the authors as supplemental material, and determined which ATAC regions were differentially enriched in only a single cell type and overlapped these with our GREs. A GRE was considered cell-type-specific if it overlapped ATAC regions differentially enriched in only a single cell type, otherwise it was annotated as not cell-type-specific.

5) Fig 4a shows significant overlap between Hominin gains and ASD up as well as ASD down. The authors only focus on ASD down in Fig4f. It is not clear why they don't show ASD down up too.

We do not show a significant overlap between hominin gains and both ASD up and ASD down regions. We find a significant overlap between gains and ASD down regions, while ASD up are significant depleted from these regions as indicated by the grey color.

6) What do the results (especially Fig2 and 3) look like when you include - or separately perform it with- hominin losses ? This could at least mentioned in the text since the authors claim that losses are more likely to be biologically relevant and selected against.

We think that losses occur at a lower rate compared to gains as due to their conservation between marmoset and rhesus (which losses need to be). Given this conservation these are more likely to have a biological function and their loss may therefore be selected against. Thus, the ones that we do pick up in our analysis are not more likely to be biologically relevant but rather the ones that have little impact. We have modified this sentence in the manuscript to make our intended message clearer.

The reason why we do not focus on losses is as we did not observe them overlapping significantly with ASD regions or enrich for a cell type or developmental stage (See panel a and b of figure below). This is expected as losses are simply not there, making cell-type or temporal assessment in the available human data not relevant. As expected, genes linked to hominin-specific losses are expressed at a lower level in PFC (Supplementary Fig. 7d) as well as in WM samples (panel c of figure below).

Hominin-specific losses. **a)** metaplot analysis showing the average H3K27ac enrichment in different cell types for hominin-specific PFC losses. **b)** box plots showing normalized H3K27ac enrichment in prefrontal cortex tissue across different developmental timepoints for all GREs, hominin-specific gains and losses. Dissimilarity between distributions were calculated using a Student's *t*-test. **c)** box plots showing the normalized gene expression values in white matter tissue of different primates for genes linked to hominin-specific PFC losses. Dissimilarities between the distributions were calculated using a Student's *t*-test. Bottom and top of all boxplots are the first and third quartile. The line within the boxes represents the median and whiskers denote interval within 1.5x interquartile range from the median, outlier points are depicted as points.

Reviewer #2 (Remarks to the Author):

I appreciate that authors have addressed a lot of my comments and added substantial amount of work during the revision. My major question on the batch correction has been resolved, and the link to oligodendrocytes is now robust. However, the link to ASD is still not convincing, as the majority of orthogonal validation neither yielded consistent nor significant results. Currently the only support they got related to ASD was the overlap with differential H3K27ac in ASD postmortem brains, while I am not convinced that this necessary represents any causal link to ASD.

Given that authors used c-MET locus as an example of the ASD locus, this gene was neither captured from the largest ASD exome sequencing (Satterstrom et al., bioRxiv; Ruzzo et al., Cell) nor GWAS (Grove et al., Nat Genet). I am wondering whether they can find additional evidence with other ASD risk genes. They have named a few, but none of them seems to be the high-confident candidate ASD genes.

We do not claim a direct causal link, just that a set of elements that recently evolved is preferentially affected in ASD, which we believe this is a fair statement. We do now omit the use of the term high confidence for regions that need further examination such as c-met and also show different risk genes including CAMK2A and DNMT3A. Overlaying our data with recently identified ASD risk genes, we find

that a hominin-specific GRE that was also shown to be deregulated in ASD by Sum W et al., was linked by HiC to DNMT3A, a gene identified as ASD risk gene in both manuscripts suggested by the reviewer (Satterstrom et al., bioRxiv; Ruzzo et al., Cell). We added this in Fig. 4b, c and Supplementary Fig. 8c, d (see also figure below) and mention DNMT3A in the text. Moreover, CAMK2A which was already provided as example in our manuscript is identified as risk gene by Ruzzo et al., We now cite these manuscripts.

Hominin-specific PFC gain linked to DNMT3A. a) ChIP-seq track showing rpm normalized H3K27ac enrichment across a 540kb regions containing the DNMT3A gene and a hominin-specific PFC gains that is deregulated in autism (highlighted in green). Chromosomal interactions, as defined by HiC in PFC, are depicted below. **b)** ChIP-seq tracks showing rpm normalized H3K27ac enrichment across the same regions as in **a** but for prefrontal cortex tissue of different primate species. Hominin-specific PFC gains highlighted in green.

Reviewer #3 (Remarks to the Author):

I'm happy with authors' response to my comments. I have no additional questions or comments.

Reviewers' Comments:

Reviewer #1:

Remarks to the Author:

The authors have reasonably addressed my further concerns.

Reviewer #2:

Remarks to the Author:

The authors have thoroughly addressed the points raised in my review, and have improved the paper.